# Associations between Objectively Determined Physical Activity and Cardiometabolic Health in Adult Women: A Systematic Review and Meta-Analysis

**DOI:** 10.3390/biology11060925

**Published:** 2022-06-17

**Authors:** Yining Lu, Huw D. Wiltshire, Julien S. Baker, Qiaojun Wang, Shanshan Ying, Jianshe Li, Yichen Lu

**Affiliations:** 1Faculty of Sport Science, Ningbo University, Ningbo 315000, China; st20184530@outlook.cardiffmet.ac.uk (Y.L.); yingshanshan@nbu.edu.cn (S.Y.); lijianshe@nbu.edu.cn (J.L.); 2Cardiff School of Sport and Health Sciences, Cardiff Metropolitan University, Cardiff CF52YB, UK; hwiltshire@cardiffmet.ac.uk; 3Centre for Health and Exercise Science Research, Department of Sport, Physical Education and Health, Hong Kong Baptist University, Kowloon Tong, Hong Kong; jsbaker@hkbu.edu.hk; 4Department of Sport and Physical Education, Zhejiang Pharmaceutical College, Ningbo 315000, China; luyc@mail.zjpc.net

**Keywords:** accelerometer, pedometer, physical activity, steps, cardiometabolic health, adult women

## Abstract

**Simple Summary:**

This systematic review and meta-analysis aimed to investigate the association between objectively measured physical activity and cardiometabolic health in adult women. After searching four databases (PubMed, Web of Science, Scopus, and the Cochrane library), 23 eligible studies were included (n = 2105). An accelerometer or pedometer determined physical activities (daily steps, total physical activity, minutes engaged in physical activities at different intensities, and the number of physical activity bouts) and cardiometabolic health indicators (blood pressure, lipids, carbohydrate metabolism, insulin, inflammation markers, and metabolic syndrome) were examined in adult women. Overall, it is compelling that being more physically active has favorable effects on the metabolic syndrome. However, the majority of individual cardiometabolic biomarkers hardly improved following increases in physical activity, with the exception that moderate-intensity physical activity appeared to have a more potent effect on high-density lipoprotein. Although higher-intensity physical activity is more effective for women, it is most important to increase the total volume of physical activity. Meanwhile, strategies to improve body composition and cardiorespiratory fitness are required, since these play an important role in mediating the association between physical activity and cardiometabolic health in women.

**Abstract:**

The purpose of this systematic review and meta-analysis was to qualitatively synthesize and quantitatively assess the evidence of the relationship between objectively determined volumes of physical activity (PA) and cardiometabolic health in women. Four databases (PubMed, Web of Science, Scopus, and the Cochrane library) were searched and, finally, 24 eligible studies were included, with a total of 2105 women from eight countries. A correlational meta-analysis shows that moderate-to-vigorous intensity physical activity (MVPA) was favorably associated with high-density lipoprotein (r = 0.16; 95% CI: 0.06, 0.25; *p* = 0.002); however, there was limited evidence for the effects of most of the other cardiometabolic biomarkers recorded from steps, total physical activity, light- and moderate-intensity physical activity and MVPA. It is most compelling and consistent that being more physically active is beneficial to the metabolic syndrome. Overall, PA levels are low in adult women, suggesting that increasing the total volume of PA is more important than emphasizing the intensity and duration of PA. The findings also indicate that, according to the confounding effects of body composition and cardiorespiratory fitness, meeting the minimal level of 150 min of moderate-intensity physical activity recommended is not enough to obtain a significant improvement in cardiometabolic indicators. Nonetheless, the high heterogeneity between studies inhibits robust conclusions.

## 1. Introduction

The rising prevalence of physical inactivity is one of the greatest public health concerns. Physical inactivity has been epidemiologically evidenced to be associated with cardiovascular diseases (CVD), which remains the leading cause of mortality [1,2]. Since physical inactivity is among the major modifiable risk factors for CVD, there is a growing need for the promotion of physical activity (PA). The latest PA guidelines recommend at least 150–300 min of moderate-intensity physical activity (MPA) or 75–150 min of vigorous-intensity physical activity (VPA) a week for adults to maintain cardiometabolic health [3,4]. Substantial observational evidence suggests that a higher level of PA is associated with a lower risk of CVD [4,5,6,7]. Intervention studies also report favorable changes in cardiometabolic risk factors after exercise interventions [8,9,10].

However, most of the evidence comes from moderate-to-vigorous-intensity physical activity (MVPA). Moreover, the global and national guidelines on PA specify recommended volumes for MPA, VPA and MVPA, with little consideration for other patterns of PA, such as light-intensity physical activity (LPA), total physical activity (TPA) and the number of daily steps. Emerging evidence indicates a dose–response relationship between total physical activity (TPA) and the incidence of CVD [11,12,13]. Furthermore, a recent meta-analysis reported beneficial effects of LPA on cardiometabolic health [14]. Despite the health benefits of PA, 27.5% of adults fail to follow the lowest level of recommended PA, and women are more physically inactive, with 30.7% inactivity in women compared to 23.4% inactivity in men [15]. Although the recommended PA is the same for both genders, women have specific anatomical, hormonal, and cardiovascular features, suggesting gender differences in the risk factors and management of CVD. For example, women have a smaller vessel size than men and suffer from a higher age-related risk of hypertension, especially postmenopausal women with decreased estrogen [16]. Furthermore, physical inactivity in women is more likely to be diagnosed with obesity [17,18]. Therefore, there is a need to explore the association between PA and cardiometabolic health in women.

Furthermore, previous studies and PA guidelines mostly relied on the PA questionnaire, which is less accurate in women [19] and struggles to provide precise measures of LPA [20]. Although objective measures of PA have been widely used, there is a paucity of evidence on the associations between objectively determined PA and clinically relevant cardiometabolic biomarkers in healthy adult women. Therefore, the purpose of this systematic review and meta-analysis is to qualitatively synthesize and quantitatively assess the association between objectively determined PA and cardiometabolic health in adult women.

## 2. Methods

This systematic review and meta-analysis were conducted following the Preferred Reporting Items for Systematic Reviews and Meta-Analyses (PRISMA) [21].

### 2.1. Inclusion Criteria and Study Selection

Participants, interventions, comparisons, and outcomes (PICO)-formatted research questions [22] were used to clarify the inclusion criteria.

#### 2.1.1. Participants

Participants included apparently healthy women with a mean age of 18–64 years. Women with the presence of cardiovascular disease risk factors (e.g., overweight/obesity, hypertension, elevated fasting glucose, and dyslipidemia) were included. Exclusion criteria included: (1) diagnosed CVD, diabetes, physical or psychological disorders, or other conditions that were barriers to physical activities; (2) pregnant, postpartum, or lactating women; (3) elite athletes.

#### 2.1.2. Interventions

Accelerometer- and pedometer-assessed PA volume was identified using interventions, including steps, counts, the LPA, MPA, VPA, MVPA, PA bouts, and TPA. Energy expenditure was not included because the accelerometer-derived data showed poor accuracy in estimating energy expenditure [23].

#### 2.1.3. Comparisons

Various steps and volumes of objectively measured physical activities were identified as comparisons.

#### 2.1.4. Outcomes

According to the literature review, both traditional cardiometabolic risk factors and novel CVD biomarkers were included. These indicators were classified into six categories: (1) blood pressure (BP) (2) lipid profile; (3) carbohydrate metabolism; (4) endocrine regulators; (5) inflammation marker; (6) metabolic syndrome (MS).

### 2.2. Study Design

Both observational (cross-sectional and longitudinal) and experimental (randomized and non-randomized) studies investigating the relationship between accelerometer- or pedometer-measured PA and cardiometabolic health biomarkers were included.

### 2.3. Other Criteria

Only original articles published in English and in a peer-reviewed journal were included. Reviews, abstracts, conference proceedings, and short reports were all excluded. Furthermore, studies focusing on cardiac rehabilitation and secondary CVD prevention programs were excluded. Studies included participants of both genders and were only eligible for inclusion when separate data for women were available. When more than one article was from the same study, the following hierarchy was applied for inclusion: (1) the largest sample size, (2) the longest following period, and (3) the most detailed data.

### 2.4. Literature Search

Four electronic databases, including PubMed, Web of Science, Scopus, and the Cochrane Library, were searched from 1 January 1990 to 31 January 2022 in accordance with the search strategy developed by two researchers with expertise in systematic reviews. Firstly, keywords such as “accelerometer”, “pedometer”, “objectively”, “physical activity”, and terms of CVD biomarkers were applied to titles and abstracts since there was no standardized keyword to fully capture the studies, including women-specified associations. Secondly, we conducted a manual search to screen the full text for eligible studies. Thirdly, the reference list from included studies was manually screened to ensure completeness of records. Finally, search results were all imported in Endnote (Endnote 20, Wintertree Software Inc., Beijing, China). The detailed search strategy is provided in Appendix B.

### 2.5. Data Extraction

For each included study, descriptive data, intervention, and correlational findings were independently extracted by two reviewers (Yining Lu and Qiaojun Wang) and inputted into Excel (Microsoft Corp, Redmond, WA, USA). Any disagreements were resolved through discussion and all results were checked by a third reviewer (Shanshan Ying). The relationship between PA and cardiometabolic health outcomes was included if it was measured by *t*-test/Mann–Whitney U-test (U-test)/Kolmogorov–Smirnov test (K-S test), analysis of variance (ANOVA), correlation, regression, and relative risks.

### 2.6. Risk of Bias and Quality Assessment

The Newcastle–Ottawa Scale (NOS) was used to assess the risk of bias in nonrandomized studies (non-randomized interventions and observational studies) [24]. For the comparability domain, we considered age to be the most important confounder. The maximum number of stars was seven for the cross-sectional design and nine for the longitudinal design. High quality was defined as four or more stars in cross-sectional designs, and five or more in longitudinal designs. Those below the cut-off point were defined as low quality [25]. The Cochrane collaboration tool was used to assess the risk of bias for random experiments [26].

The Grading of Recommendations Assessment, Development, and Evaluation (GRADE) was used to evaluate the quality of evidence for each category of biomarkers [27]. 

Each included study was independently rated by two reviewers (Yining Lu and Shanshan Ying). Any discrepancy in rating was resolved by discussion and the results were checked by a third reviewer. Details on the risk of bias and quality assessments are presented in Appendix A.

### 2.7. Statistical Analysis

Comparable PA exposures included steps, minutes in LPA, MPA, VPA, MVPA and PA bouts, TPA, meeting/not meeting the guideline. Although different cut-off points and definitions were used, LPA, MPA, VPA, MVPA and TPA were defined as reported in the studies [28]. If studies measured physical activity in metabolic-equivalent tasks (METs), we used the cut-points proposed by Ainsworth et al. (2011) (e.g., 1.6–2.9 METs was defined as LPA, 3–5.9 METs as MPA, and ≥6 METs as VPA) [29]. When more than one statistical analysis was used, the following hierarchy was applied: (1) regression, (2) correlation, (3) ANOVA, (4) *t*-test/U-test/K-S test [25]. When more than one adjusted model was used, the most-adjusted models were applied [30].

Meta-analysis was planned if more than two studies were eligible for comparable PA measures and biomarkers. Fisher’s z transformation and Hedge’s g was used for correlational and standardized mean differences meta-analysis, respectively [31]. The effect size was classified as low (r = 0.1/SMD = 0.2), moderate (r = 0.3/SMD = 0.5), or high (r = 0.50/SMD = 0.8) according to Cohen’s recommendations [32].

The random-effects model was used due to the diversity of the methodologies. To evaluate the impact of heterogeneity on the meta-analysis, inconsistency was measured using the Higgins’ I^2^ statistic. Specifically, I^2^ = 0 indicated no heterogeneity and low, moderate and high heterogeneity was identified when I^2^ < 25%, 25–75%, and >75%, respectively [33]. Publication bias was assessed through Egger’s test and funnel plots using at least ten studies [34]. Subgroup analyses were conducted to examine the potential sources of heterogeneity, including age (young (18–39 years old)/middle-age (40–64 years old)/both) [35], BMI (BMI < 25/BMI ≥ 25), menopausal status (postmenopausal/premenopausal/both), country, and ethnicity. All statistical analyses were performed with Review Manager, version 5.4.1 (The Cochrane Collaboration, London, UK, 2020).

## 3. Results

### 3.1. Study Selection and Characteristics

A total of 5112 records were yielded from the database and manual search. After screening 126 full texts, 23 eligible studies were finally included in the precent review. The most common reason for exclusion was the unavailability of female-specific data. Figure 1 presents the PRISMA system outlining the study process.

The characteristics of the observational and intervention studies that were included are detailed in Table 1 and Table 2, respectively. Publication dates ranged from 2001 to 2021. Out of 23 studies, 14 studies were cross-sectional designs [36,37,38,39,40,41,42,43,44,45,46,47,48,49] and 9 were interventional studies [50,51,52,53,54,55,56,57,58] (random experiment: 5 [51,52,54,56,57] and non-random experiment: 4 [50,53,55,58]). Intervention length ranged from 1 week [55] to 24 months [37].

### 3.2. Sample Characteristics

The total sample size was 2105, ranging from 10 [55] to 535 [42] participants. The mean age ranged from 21.4 [50] to 62.8 years [49], with 8 studies [36,38,39,43,45,47,48,50] focused on the young and 10 studies focused on the middle-aged [37,41,42,44,49,51,52,56,57,58]. There were 11 studies that reported menstrual status, with 4 including post-menstruation [37,49,52,56], 5 including pre-menstruation [38,39,41,43,45], and 2 including both [57,58]. Moreover, there were 11 studies focused on overweight/obese females [36,37,40,42,43,44,51,52,54,55,58]; 10 studies included physically inactive participants [37,46,51,52,53,54,55,56,57,58]; 12 studies reported smoking status, with 5 including non-smokers [41,52,54,56,58], 4 reporting on participants that refrained from tobacco in the last 6 months [39,47,48,51], and 3 studies including mostly non-smokers (60–83%) [36,42,44]. Education level was reported in 4 studies [39,40,41,50], and social-economic levels were presented as low-income in 4 studies [40,43,44,45].

The included studies were conducted in eight countries. There were 3 studies [50,56,57] that included a sample from Asia (i.e., Japan and UAE), 2 from Europe (i.e., Italy and Poland), 13 studies [36,39,40,41,42,44,47,48,51,52,53,55,58] from North America (i.e., USA), 3 studies [37,38,43] from South America (i.e., Brazil), and 2 studies [45,54] from Oceania (i.e., New Zealand and Australia). The race reported in the studies included Caucasian, Asian, Latina, African American, Hispanic, Pacific, Mexican American, European. See Table 3 for details of sample characteristics.

### 3.3. Physical Activity Assessment

Accelerometers were used in 12 (85.7%) observational studies [36,37,39,40,41,42,43,45,46,47,48,49] and 2 (22.2%) experimental studies [55,56]. Two (14.3%) observational studies [38,44] and seven (77.8%) experimental studies used pedometers [50,51,52,53,54,57,58].

From 14 observational studies, objectively assessed PA included minutes in LPA, MPA, VPA, MVPA and PA bouts, steps, TPA, and met or did not meet the PA guidelines. Steps were assessed in 7 (77.8%) interventional studies [50,51,52,53,54,57,58]. The PA intensity was categorized using various cut-off points, including accelerometer counts and metabolic equivalents (METs). Detailed ascertainment and measurement characteristics of objectively measured PA are illustrated in Appendix C.

### 3.4. Cardiometabolic Health Outcomes Assessment

Cardiometabolic biomarkers assessed in the studies included BP (systolic blood pressure (SBP), diastolic blood pressure (DBP)), lipid profile (total cholesterol (TC), high-density lipoprotein (HDL), low-density lipoprotein (LDL), triglyceride (TG)), carbohydrate metabolism (fasting glucose (FPG), postprandial glucose (PPG), glycosylated hemoglobin (HbA1c), homeostasis model assessment of insulin resistance (HOMA-IR)), endocrine regulators (fasting insulin, postprandial insulin), inflammation marker (C-reactive protein (CRP), interleukin-6 (IL-6), and tumor necrosis factor-α (TNF-alpha) and MS.

### 3.5. Risk of Bias Assessment and the Quality of Evidence

The risk of bias assessment for the included studies is presented in Appendix A. Out of 18 observational or non-randomized designs, 13 were categorized as high-quality [36,37,39,40,41,42,43,45,46,47,48,49,50] and 5 of low-quality [38,44,53,55,58]. Ten (55.6%) of the studies did not control for age [38,39,40,44,47,49,50,53,55,58], which was the most important covariate that we determined for quality assessment. The majority of random experiments were of unclear quality (80%), with only one of high quality [51]. The most-reported risk of bias came from the lack of random sequence generation.

Moreover, according to the GRADE framework, very low to moderate quality evidence was reported, with none being upgraded. The small sample size of the intervention studies was the most common reason for the downgrade, and the lower quality for observational studies was mostly due to the inconsistency of findings. Appendix A presents the details on the quality of evidence according to study design and the categories of cardiometabolic health outcomes.

### 3.6. Association between PA and Cardiometabolic Health Outcomes

From 23 studies, 13 studies assessed the association between PA and BP [39,40,44,45,47,48,50,51,52,53,54,56,58], 11 studies assessed lipid profile [38,39,40,44,45,47,48,50,51,53,57], 16 assessed carbohydrate metabolisms [37,38,39,40,41,43,44,45,46,47,48,50,51,52,53,58], 8 assessed endocrine regulators [37,38,39,45,47,50,52,58], 6 assessed inflammation markers [37,39,44,45,47,51], and 4 assessed MS [36,42,49,50,51].

#### 3.6.1. Blood Pressure

Observational studies reported consistent findings that there was no relationship between PA and BP [39,40,44,45,47,48]. However, intervention studies found some favorable effects. One of the four random experiments reported improvements in SBP with an average of 9700 steps/d across the 24-week walking program [52]. Out of three quasi-experimental studies, one indicated significantly decreased SBP and DBP after the 8-week intervention, during which an average of 85% increased to 9213 steps per day [58]. Another reported that only SBP was significantly improved after a 12-week incremental pedometer program [53].

#### 3.6.2. Lipid Profile

Both observational and intervention studies reported some favorable relationships with lipid outcomes. One quasi-experimental study found a favorable effect on LDL after a 9-week walking program with 7056 average daily steps [50]. However, from three random experiments, only one study conducted a 24-month moderate-intensity exercise intervention, reporting favorable effects on HDL and TC [57]. The other two random experiments studies showed no effect of increased daily steps on HDL, TG, or TC [51,53].

From nine studies that reported cross-sectional evidence, three studies (75%) reported favorable associations between daily steps and HDL [48] and TG [40,48]. Furthermore, LPA [39] and meeting the recommended 150 min MVPA weekly [47] were reported to be beneficial to TG and TC. Two studies (66.7%) suggested a favorable association between MVPA and HDL [40,45]. Of note, an unfavorable but small relationship with TC was found in overweight Latin women [40]. Similarly, unfavorable relationships with LDL or TC were reported in obese African American women with a lower socioeconomic status [44].

#### 3.6.3. Carbohydrate Metabolism

Observational studies reported consistent findings that there was no association between PA and FPG or HbA1c [39,40,43,44,45,46,47,48]. HOMA-IR was found to be favorably associated with daily steps [38] and MVPA bouts [39]. Similarly, there was a favorable association between MVPA and peak PPG [46].

Although two random experiments reported no effects on FPG or HbA1c, which was consistent with findings from observational studies, HOMA-IR was improved as a result of engaging in a walking program [51,52]. These findings were not replicated in quasi-experimental studies as a favorable effect on FPG [53], and PPG [58] was observed following walking programs. In addition, one study used a crossover design to examine the effects of three conditions on carbohydrate metabolism and found that increasing the percentage volume of MVPA had a favorable effect on PPG, while LPA had no effect [55].

#### 3.6.4. Endocrine Regulation

One randomized controlled trial and two quasi-experimental studies consistently indicated no effects on fasting insulin or postprandial insulin after engaging in the walking program [50,52,58]. However, some favorable associations were found in cross-sectional observational studies. One study found favorable associations between daily steps with fasting insulin and postprandial insulin [38]. Furthermore, fasting insulin was shown to be favorably associated with 10-min MVPA bouts [39]. It was worth noting that one study reported an unfavorable association between TPA and fasting insulin [45].

#### 3.6.5. Inflammation Markers

Only one random experiment examined the effect on inflammation markers and reported that increasing daily steps had no effect on CRP after a 12-week exercise intervention [51]. However, one observational study reported favorable associations between CRP and MVPA and MVPA bouts [39]. The opposite results were reported by Slater et al. (2021), who found that both TPA and MVPA were detrimental to CRP [45]. There was no relationship between PA and TNF-alpha [37] or IL-6 [39].

#### 3.6.6. Metabolic Syndrome

Findings regarding to MS were unequivocal in intervention and observational studies. One quasi-experimental study reported a positive effect on MS score after participating a 9-week walking program [50]. Cross-sectional studies found the incidence of MS was favorably associated with daily steps [49], the volume of LPA [36] and MVPA [42].

### 3.7. Meta-Analysis

There were six studies (4 cross-sectional, 1 quasi-experimental and 1 RCT) that provided correlational data, which could be pooled to conduct the meta-analysis. Table 4 and Figure 2 illustrate the correlational meta-analysis for the included studies.

The results from three studies assessed MVPA could be pooled into a meta-analysis (n = 545). The pooled results showed a significantly favorable but small relationship between MVPA and HDL (r = 0.16; 95% CI: 0.06, 0.25; *p* < 0.01), with a low heterogeneity between studies (I^2^ = 19%, *p* = 0.29). However, according to subgroup analysis, no significant relationship between MVPA and HDL was detected in young (n = 2) and premenopausal (n = 2) women. Studies conducted in the US (n = 2) and Caucasian (n = 1) women also showed no relationship between MVPA and HDL.

In contrast, there was a pooled unfavorable but small relationship between MVPA and TC (r = 0.09; 95% CI: 0.00, 0.18; *p* < 0.05), with no between-study heterogeneity (I^2^ = 0%, *p* = 0.40). The subgroup analysis showed that the unfavorable association could be explained by obesity (n = 1) and Latin ethnicity (n = 1).

According to a pooled analysis, there were not any significant associations between MVPA and DBP, SBP, LDL, FPG, and TG. Subgroup analysis revealed that age, BMI, menstrual status, country, and ethnicity had no effect on the association.

There was no significant correlation between daily steps and FPG (n = 313) (r = −0.12; 95% CI: −0.24, 0.01; *p* = 0.06, n = 3). The between-study heterogeneity was low (I^2^ = 42%, *p* = 0.18). The subgroup analysis based on the age, BMI, menstrual status, country, and ethnic did not modify the association.

The pooled analysis of four studies (n = 349) revealed that daily steps were not associated with HDL (r = 0.24; 95% CI: −0.07, 0.54; *p* = 0.13), with a high heterogeneity between studies (I^2^ = 84%, *p* < 0.05). However, subgroup analysis (n = 14) revealed a significantly stronger association in middle-aged women in studies conducted in Japan (r = 0.85; 95% CI: 0.58, 0.95; *p* < 0.001; n = 1). The between-study heterogeneity was mostly explained by the subgroup analysis of BMI.

Additionally, we conducted a meta-analysis to examine the effect of meeting PA guidelines on HOMA-IR, since there were three comparable studies (Table 5 and Figure 3). The mean and standard deviation were extracted from those who met the 150 min of MPA and those who did not to calculate the standardized mean differences. The pooled results showed that following the recommended MVPA level had no significant effect on HOMA-IR (SMD= −0.22; 95% CI: −0.46, 0.02; *p* = 0.08), with a low heterogeneity (I^2^ = 11%, *p* = 0.32). According to subgroup analysis, meeting PA guidelines was significantly associated with lower HOMA-IR in studies conducted in the USA (n = 2) and in Caucasian women (n = 1).

## 4. Discussion

This systematic review and meta-analysis are the first to synthesize studies that investigated the association between objectively determined PA volume and clinically relevant cardiometabolic biomarkers in adult women across a range of study designs. Although relatively limited by the small number of included studies, the evidence examining the association between objectively assessed PA and cardiometabolic indicators points towards a favorable association between MVPA and HDL. Evidence of a beneficial effect on other cardiometabolic outcomes seems to be limited.

### 4.1. Meta-Analytic Findings

Findings from the meta-analysis revealed that spending more minutes on MVPA were significantly associated with a healthier HDL; however, the pooled effect size was small. No significant associations were observed between MVPA and most cardiometabolic biomarkers, including SBP, DBP, LDL, FPG, and TG.

Subgroup analysis showed significant differences in the association between steps and HDL across various ages, countries and ethnicities. However, it was noteworthy that the number of studies included in the subgroup was small, and caution is required when drawing conclusions from subgroup analyses.

### 4.2. Association between Steps and Cardiometabolic Biomarkers

The observational and experimental evidence examining the associations between daily steps and cardiometabolic health indicate no effects on improving cardiometabolic biomarkers, including BP, lipids, glucose, insulin, and inflammation markers. This was supported by the meta-analytic findings that daily steps were not significantly associated with HDL or FPG.

Even though most studies reported no association between steps and BP, three studies conducted a long-term walking program in obese women and observed decreases in SBP after the intervention [52,53,58]. Participants in these three studies were all obese, with elevated or stage I high SBP at baseline, and moreover, their daily steps doubled to about 10,000 after the intervention. This favorable effect was supported by a systematic review, in which the decreased SBP was found to be associated with higher baseline values and the magnitude of change in steps per day [59].

Experimental evidence suggests that increasing daily steps after intervention had no effects on HDL, LDL, TG or TC [51,53]. A pedometer-based walking intervention reported no effects on HDL, TG, or TC, while LDL improved at the end of the intervention [50]. The improved LDL was mainly due to weight loss and improved body composition [60]. However, one intervention study reported favorable associations between steps with HDL and TC after the intervention [57]. It was supposed that the improved lipid profiles were mostly due to additional moderate-intensity aerobic training rather than the increased steps, since aerobic exercise interventions had a more consistent and potent effect on improving HDL and associated cardiometabolic health indicators [61,62].

Cross-sectional evidence revealed that there was no association between steps and most blood lipids [38,40,44,48]. This was supported by a cross-sectional study, in which no differences were observed for TC, LDL, HDL, and TG between the group with more than 7500 steps per day and the group with less than 7500 [63]. However, body composition affected this relationship, as steps were significantly associated with HDL and TG after adjusting for fat mass and fat-free mass [48]. Interestingly, the opposite findings were reported by Panton et al. (2007), who found that women who walked at least 5000 steps per day had worse LDL and TC than those walking less than 5000 steps [44]. A potential explanation was that, for obese women, more steps were needed to improve lipid markers.

There was consistent evidence from experimental and observational studies that steps had no effects on carbohydrate metabolism [38,40,46,48,50,51,52]. Results from other intervention studies also supported the idea that walking programs had no effect on improving FPG [64,65]. Although most studies reported no improvements in FPG, a favorable effect on PPG was observed after increasing daily steps during the 8-week walking program [58]. This finding was consistent with a prospective study that there was a weak favorable correlation between previous daily steps and 2 h-PPG, but no correlation with FPG [65].

Experimental studies reported consistent findings that increased daily steps after intervention had no effects on insulin sensitivity [50,52,58], while a cross-sectional study observed the lower levels of fasting insulin and HOMA-IR in more active women [38].

Despite there being no relationship between daily steps and cardiometabolic health outcomes suggested by most of the studies, consistent beneficial effects were observed for MS score, which was defined as the sum of the number of individual MS indicators [49,50]. Both observational and intervention studies found a favorable association between steps and MS score, which could be supported by longitudinal studies. Huffman et al. (2014) conducted an observational study from NAVIGATOR and found that baseline steps were independently associated with reductions in MS score, which was calculated by summing each standardized MS component [64]. Ponsonby et al. (2011) followed 458 adults with normal glucose and found that a higher level of daily steps was associated with a lower risk of the incidence of abnormal glucose metabolism 5 years later [66].

Walking is incidental to daily life and the accumulated number of steps were mostly at a low intensity. Walking intensity was more important than walking volume in terms of the association with cardiometabolic health, and this might explain why the beneficial effects on cardiometabolic biomarkers were hardly observed [67]. In addition to walking intensity, evidence from experimental studies indicated that the baseline value of biomarkers and magnitude of changes in daily steps affected the relationship between walking and cardiometabolic health outcomes. Likewise, body composition variables suggested by observational cross-sectional evidence could also mediate this relationship. Therefore, more controlled experimental and prospective studies with high-quality experimental designs are needed in the future.

### 4.3. Association between TPA and Cardiometabolic Biomarkers

Evidence from the current review revealed that there was no association between TPA and most cardiometabolic markers, expect fasting insulin and CRP [45]. Among obese women, there was no significant difference in TPA between metabolically healthy and metabolically unhealthy women [36]. However, a cross-sectional finding revealed that TPA displayed stronger associations with cardiometabolic biomarkers, including HDL, TG, FPG, fasting insulin, CRP, and SBP [68]. One potential explanation for the contradictory results might be the discrepancy of TPA between genders, as women engage in less TPA than men. Additionally, the relationship varied according to the ethnicity of the subjects. TPA was associated with fasting insulin in Pacific women but not European women, and the relationship between TPA and CRP was reversed between the two ethnicities. This may be because fasting insulin was nearly two times higher in Pacific women and CRP was positively associated with visceral fat, which was higher in Pacific women.

### 4.4. Association between Volume of PA at Different Intensity and Health Outcomes

Both cross-sectional and intervention evidence supported the idea that there was no relationship between LPA and cardiometabolic health markers [39,43,55]. Our findings were consistent with a previous review [69]. This review summarized the effect of exercise protocols delivered at light intensity and showed little support for the role of LPA in improving cardiometabolic health; moreover, it indicated that the applied dose of LPA was low among the included studies. However, there was emerging evidence that LPA had benefits for health [70,71,72]. In a cross-sectional study, LPA was shown to be significantly associated with TG and TC. These associations were independent of MVPA, but were attenuated by peak oxygen uptake (VO_2peak_) and body composition outcomes, indicating that VO_2peak_ and body composition might be important contributors to cardiometabolic health [39]. Previous cross-sectional studies showed that VO_2peak_ was associated with CVD risk factors, with moderate to strong correlations [73]. Likewise, Kodama et al. (2009) conducted a meta-analysis to quantitatively define the relationship between cardiorespiratory fitness and the incidence of CVD. The authors indicated that those with low cardiorespiratory fitness had a risk ratio for CVD events of 1.56 compared to those with high cardiorespiratory fitness [74]. Therefore, the cardiorespiratory fitness appeared to be an important confounder when investigating the relationship between PA and cardiometabolic health. Furthermore, the cardiorespiratory fitness should be taken into consideration when developing exercise protocols aiming to improve cardiometabolic health. Since high-intensity exercises were well-documented to be effective and efficient in improving cardiorespiratory fitness [75,76,77,78], in this regard, PA performed at higher intensity was recommended.

Moreover, a recent systematic review identified 24 cross-sectional and 6 longitudinal studies and found that LPA appears to be independently associated with better WC, TG, fasting insulin, and the presence of MS [79]. Additionally, replacing sitting with LPA was also found to be an effective way to improve health [80]. It was plausible that there is a threshold for PA at which health outcomes improved, and the threshold for LPA would be much higher due to the lower effects accumulated by the low intensities. This statement is supported by findings from a recent systematic review examining the relationship between LPA and cardiometabolic health and mortality in adults. The authors pointed toward the beneficial effects of LPA; however, LPA effects were from two to four times lower than MVPA effects for the same duration [14]. Moreover, the current PA guidelines recommended that at least 150–300 min/week of MPA were required to observe the benefits [3]. Studies that investigated MPA exclusively were sparse. Limited evidence from the present review supported the idea that MPA had no association with HOMA-IR [41,43].

It was generally believed that MVPA appeared to be more potently associated with cardiometabolic biomarkers. However, cross-sectional evidence from the present review suggests that there are no associations between MVPA and cardiometabolic risk indicators [39,40,45,46]. Only one study found a favorable association with the odds of being MS after controlling for age, ethnicity, and smoking [42]. In this study, MS was defined if participants meet three or more of the following criteria: (1) WC ≥ 88 cm; (2) TG ≥ 150 mg/dL or self-reporting on treatment; (3) HDL < 50 mg/dL or self-reporting on treatment; (4) SBP ≥ 130 mm Hg and DBP ≥ 85 mm Hg or self-reporting on treatment); (5) FPG ≥ 100 mg/dL or self-reporting on treatment). Cross-sectional evidence suggested that effects on cardiometabolic health seemed to be limited, while some prospective studies reported beneficial associations. A 10-year longitudinal study investigated the independent association of changes in MVPA and objectively measured cardiometabolic health and concluded that a greater decrease in MVPA was associated with a greater decrease in HDL and increases in clustered cardiometabolic risk score [81]. However, MVPA was self-reported. Mielke et al. (2021) investigated the prospective association between accelerometer-determined MVPA and cardiometabolic health in the transition to adulthood [82]. The authors suggested that young women who increased MVPA from 18 to 22 years old showed improvements in cardiometabolic health at age 22, and moreover, MVPA in 10-min bouts showed a stronger interaction than MVPA in 1 min. Similarly, a previous study conducted by Strath et al. (2008) analyzed data from the 2003–2004 National Health and Nutrition Examination Survey and found that the bouts of MVPA appeared to be a time-efficient strategy [83]. However, evidence from qualitative synthesis in the current review pointed towards there being no differences in beneficial effects between bouts of MVPA that lasted for more than 10 consecutive minutes and no bouts of MVPA [39,40]. One potential explanation for this was that women often engaged in short bouts of MVPA, which were normally less than 10 min [40]. Consistent findings were also reported by recent cross-sectional research showing that the impact of accumulated PA obtained from several short bouts of exercise is the same as the benefits obtained from longer-duration activities [84,85,86,87]. These results were in agreement with findings from prospective studies that short spurts of MVPA could provide protection against the onset of hypertension [88] and all-cause mortality [89]. Although MVPA in 10-min bouts was generally recommended for its health benefits, the accumulated evidence from cross-sectional and prospective studies showed that short-lived MVPA was associated with health outcomes. As such, the move towards recommending MVPA of any duration through the PA guidelines appears be a pragmatic change [3,4].

Research focusing on total volume suggested that there was emerging evidence that the total volume of PA, not the minutes accumulated in bouts, was important in relation to health [68,86,87,88]. Moreover, PA of a sufficient volume was favorably associated with cardiometabolic health, independent of PA intensity [90]. In a cross-sectional study assessing the relationship between PA and cardiometabolic health in overweight Latina women, minutes of MVPA bouts were shorter than overall minutes of MVPA and moreover, the effect size of the correlation with cardiometabolic indicators was smaller for minutes of MVPA bouts than overall minutes of MVPA [40]. Likewise, Green et al. (2014) found that overall minutes of MVPA was a stronger variable than the bouts of MVPA regarding the association with markers of cardiometabolic health in young women [39]. Although most evidence was from cross-sectional analysis, it was encouraging that the promotion of short bouts of MVPA was more likely to be feasible for most women. From the public health perspective, this has significant implications for inactive individuals, as health benefits could be achieved by simply being more physically active without emphasizing the duration of exercise.

We also examined the effects of meeting the PA recommendation that adults should undertake at least 150 min of MPA a week; overall, meeting PA recommendations had an unclear impact on cardiometabolic health. Few significant differences were found between women who were meeting the recommendations and those who were below the recommended levels [37,41,47]. This was supported by the meta-analytic findings that there was no effect on HOMA-IR when meeting the recommended level. Only TG and TC were found to be improved by meeting the PA recommendations [47]. On the contrary, several previous studies based on large-scale populations showed that following the PA guidelines was strongly associated with a lower risk of cardiometabolic disease [91,92]. Discrepancies among these findings may be explained by the mediating roles of body composition variables on the relationship between PA and insulin resistance [41,93]. Other research also suggested that greater adiposity was associated with higher concentrations of inflammatory markers [94,95].

It is worth noting that most large-scale studies included self-reported PA rather than objectively measured PA, which was believed to attenuate the credibility of the findings. Furthermore, the current guidelines were developed in accordance with reviewed evidence to assess associations between PA and a set of health outcomes; however, most of the evidence was based on subjectively determined PA. Despite the limitations of our relatively low-quality evidence, the results of the current review showed some support of the idea that objectively measured PA was not beneficial to most cardiometabolic outcomes. However, most studies using subjectively determined PA consistently reported a favorable association with health outcomes [96,97,98]. This discrepancy was mainly due to the weak correlation between subjective and objective methods for assessing the intensity and duration of PA [99]. We were unable to judge which was superior because both had several limitations. Therefore, a combination of subjective and objective methods would be expected to further clarify some of the issues revealed by this study.

### 4.5. Strengths, Limitations, and Future Directions

The strengths of the current study include the use of different types of study designs and the inclusion of objectively determined PA volumes. This review was the first to analyze the evidence from different study designs both qualitatively and quantitatively and to explore the association between PA volume and clinical health indicators in adult women.

It was important to note that there were some limitations. First, most of the synthesized evidence ranged from very low to low quality. This was mainly due to the small sample sizes and concerns regarding risk of bias in the results. However, we compared low-quality evidence to high-quality evidence in the discussion, and additional high-quality and well-controlled intervention studies with a large sample size will be required to increase the confidence of the findings presented here.

Secondly, most of the included studies were cross-sectional in design, using *t*-test or ANOVA without controlling for any potential confounders, such as age and body composition. An initiative to address this issue was setting age to be the most import confounder when rating the quality of the included studies. Furthermore, the most-adjusted data were included in the discussion and meta-analysis. In addition, although sedentary behavior was documented to be associated with cardiometabolic health, it was not assessed in the current review. Taken together, the absence of these confounders attenuated the association between PA and cardiometabolic health, and our findings should be interpreted with caution.

Thirdly, our findings must be interpreted with the methodological consideration that PA at different intensities was defined as reported in the studies. Therefore, the heterogeneity in the different definition of PA categories, including different cut-points of counts, METs, and vertical acceleration peaks, was a potential source of inconsistent findings. Furthermore, the use of different epochs might also contribute to overestimation or underestimation of the amount of PA at a particular intensity. For instance, studies using longer epochs (e.g., 10 min) were more likely to underestimate the volume of higher-intensity PA than those using 60-s epochs. Finally, accelerometer-determined PA was unable to quantify certain activities, such as yoga, Pilates, and swimming, and unable to precisely calculate the energy expenditure of PA. Likewise, the pedometer was unable to quantify the intensity of walking. To deal with these limitations, standardized cut-points, shorter epochs, and pattern recognition should be applied in future.

Lastly, findings from intervention studies were synthesized with a small sample size. Further large-scale intervention studies were need. Likewise, findings from subgroup analyses were limited and should be considered preliminary due to there being only a few studies including each subgroup category.

## 5. Registration

This protocol was registered in the International Prospective Register of Systematic Reviews (PROSPERO). The registration name was physical activity and health indicator in women: a systematic review and meta-analysis, and the registration number was CRD42022307774 (https://www.crd.york.ac.uk/prospero/display_record.php?RecordID=307774). The current systematic review and meta-analysis were conducted with regard to the association between PA and cardiometabolic health.

## 6. Implications for Practice and Future Research

Our systematic review and meta-analysis found that accelerometer- and pedometer-derived PA were not associated with most individual cardiometabolic health outcomes. These findings were inconsistent with those based on subjectively measured PA. For future improvements in objective measures, the gender-specific cut-points, activity pattern recognition was shown to be more likely to improve our knowledge of the health benefits of PA.

Our review found evidence that walking programs were effective in increasing daily steps among adult women, while significant improvements in cardiometabolic indicators were hardly observed following interventions, except among obesity participants. However, some improvements in SBP were reported among obese women with a higher SBP value at baseline. Furthermore, we found that increasing PA was associated with a higher HDL; however, this favorable association was attenuated among young women. Further research should pay greater attention to potential confounders, such as age, body composition and cardiorespiratory fitness, when investigating the association between PA and cardiometabolic health in adult women.

## 7. Conclusions

The findings from the present systematic review and meta-analysis provide evidence that objectively measured PA is not associated with most cardiometabolic health outcomes in healthy adult women. However, it is most compelling that being more physically active is beneficial for MS. For women, it makes more sense to emphasize the volume of PA rather than whether the volume of PA is sporadic or occurs in bouts. Even though low-to-moderate-intensity PA contributes the most to the PA patterns observed in women, PA performed at a higher intensity is more effective in improving cardiometabolic health. The present review also highlights that meeting the recommended 150 min of MVPA each week is not enough to observe significant beneficial effects. However, further high-quality studies with less heterogeneity are needed to yield compelling findings on the association between PA and cardiometabolic health in women. 

## Figures and Tables

**Figure 1 biology-11-00925-f001:**
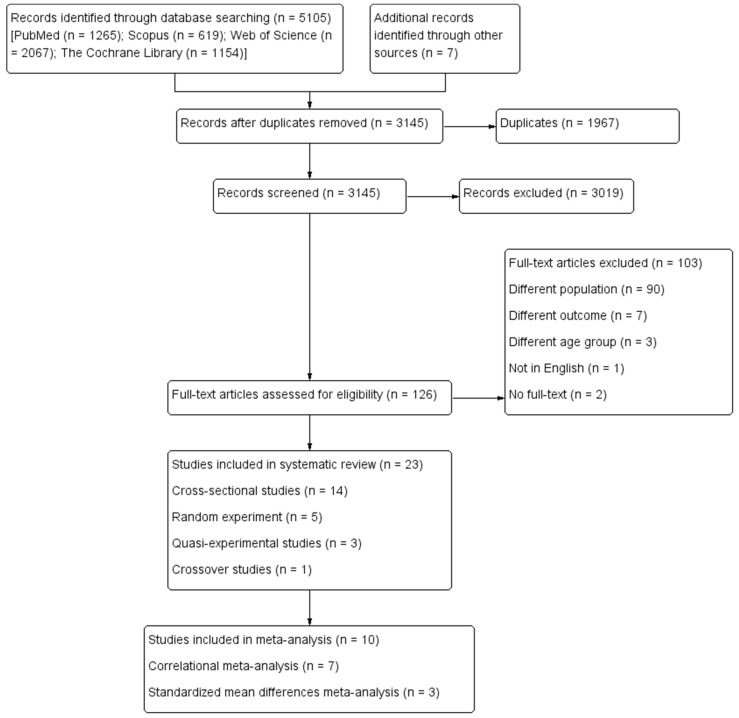
The PRISMA system for study process.

**Figure 2 biology-11-00925-f002:**
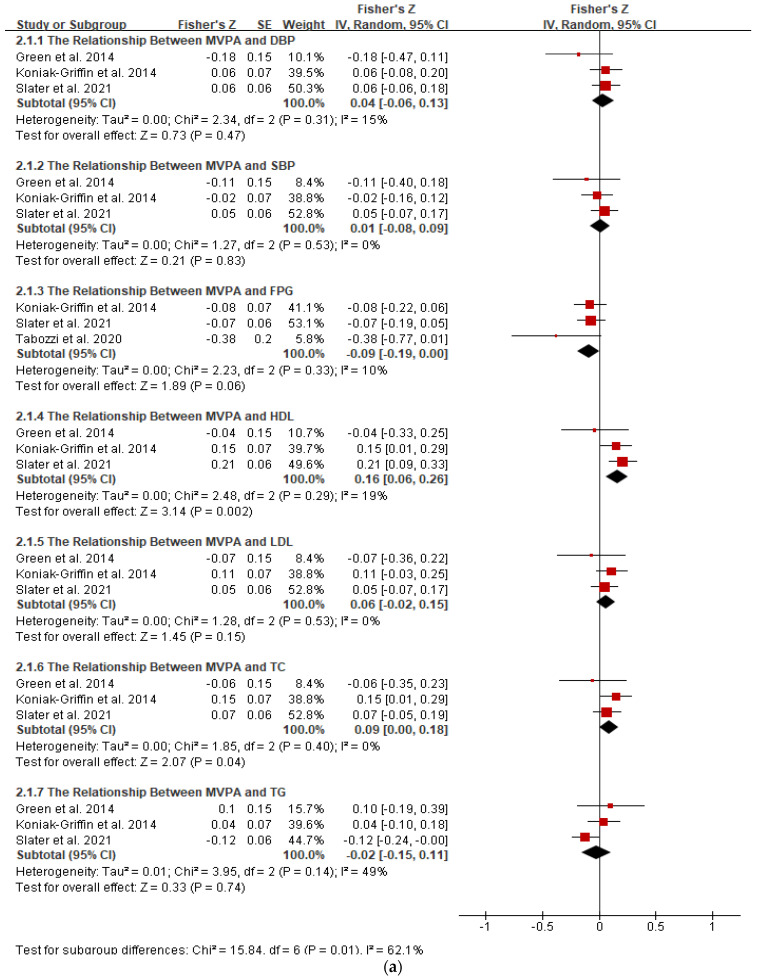
Forest plot of correlation between physical activity and cardiometabolic health outcomes. Overall pooled correlation for random effects model represented by black diamond. (**a**) The relationship between MVPA and cardiometabolic indicators. Note: DBP, diastolic blood pressure; FPG, fasting glucose; HDL, high-density lipoprotein; LDL, low-density lipoprotein; MVPA, moderate-to-vigorous-intensity physical activity; SBP, systolic blood pressure; TC, total-cholesterol; TG, triglyceride. (**b**) The relationship between steps and cardiometabolic indicators. Note: FPG, fasting glucose; HDL, high-density lipoprotein.

**Figure 3 biology-11-00925-f003:**
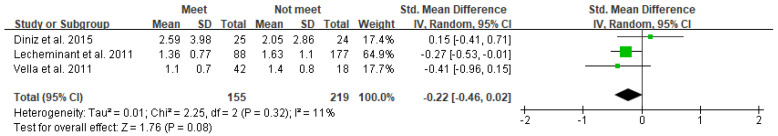
Forest plot of the effect of meeting physical activity guidelines on HOMA-IR. Overall pooled effect for random effects model represented by black diamond.

**Table 1 biology-11-00925-t001:** Characteristics of observational studies.

Reference	Study Design	Sample Size	PA Measure	Health Outcome	Association
Camhi et al., 2015 [36]	cross-sectional	46	ACC: ActiGraph GT3X+; triaxial Volume (min/d): LPA, MPA, VPA, MVPA, MVPA bout (10 min); TPA (counts/d); Steps (n/d).	MS	*t*-test: (1) MHO group had significantly higher levels of LPA compared to MUO; (2) No differences in MPA, VPA, MVPA, MVPA bouts, TPA, and steps between MHO and MUO groups.
Diniz et al., 2015 [37]	cross-sectional	49	ACC: ActiGraph GT3x; triaxial meeting/not meeting MVPA (150 min/w)	TNF-alpha, Fasting insulin, HOMA-IR	U-test: (1) Meeting MVPA volume had no effect on TNF-alpha, fasting insulin, and HOMA-IR;
Graff et al., 2012 [38]	cross-sectional	68	PED: BP 148 Steps (n/d)	TC, LDL, HDL, TG, FPG, PPG, Fasting insulin, Postprandial insulin, HOMA-IR	*t*-test and U-test: (1) No differences in TC, LDL, HDL, TG, FPG, PPG between Group (Steps/d < 6000) and Group (Steps/d ≥ 6000); (2) Group (Steps/d < 6000) had higher fasting insulin, postprandial insulin and HOMA-IR than Group (Steps/d ≥ 6000).
Green et al., 2014 [39]	cross-sectional	50	ACC: ActiGraph GT3X+; triaxial Volume (min/d) of LPA, MVPA, MVPA bout (10 min)	FPG, SBP, DBP, TG, TC, HDL, LDL, HOMA-IR, Fasting insulin, CRP, IL-6, TNF-alpha	Regression: (1) MVPA had no association with TG (adjusted for SB, VO_2peak_, BM) (2) LPA was favorably associated with TG, had no association with HOMA-IR (adjusted for MVPA, VO_2peak_, BM). Correlation: (1) LPA was favorably associated with TG, TC, HOMR-IR; had no association with FPG, SBP, DBP, HDL, LDL, fasting insulin, CRP, IL-6, TNF-alpha; (2) MVPA was favorably associated with CRP, TNF-alpha; had no association with FPG, SBP, DBP, TG, TC, HDL, LDL, HOMR-IR, fasting insulin, IL-6; (3) MVPA bouts were favorably associated with HOMA-IR, fasting insulin, CRP; had no association with FPG, SBP, DBP, TG, TC, HDL, LDL, IL-6, TNF-alpha.
Koniak-Griffin et al., 2014 [40]	cross-sectional	210	ACC: Kenz Lifecorder Plus; uniaxial Volume (min/d) of MVPA, MVPA bout (10 min) Steps (n/d)	SBP, DBP, LDL, HDL, TC, TG, FPG	Correlation: (1) Steps/d were favorably associated with TG; had no association with SBP, DBP, LDL, HDL, TC, FPG; (2) MVPA was favorably associated with HDL; unfavorably associated with TC; had no association with SBP, DBP, LDL, TG, FPG; (3) MVPA bouts had no association with SBP, DBP, LDL, HDL, TC, TG, FPG.
Lecheminant et al., 2011 [41]	cross-sectional	264	ACC: Actigraph; uniaxial Volume (min/w) of MPA, VPA meeting/not meeting MPA (150 min/w)	HOMA-IR	ANCOVA: age, weight, BMI, %BF, and ACi (1) Meeting MPA guidelines had favorable effect on HOMA-IR when adjusted for age or BM; (2) Meeting MPA guidelines had no effect on HOMA-IR when adjusted for %BF, BMI, or ACi; (3) Taking VPA ≥ 60 min/w had favorable effect on HOMA-IR when adjusted for age, BM or BMI; (4) Taking VPA ≥ 60 min/w had no effect on HOMA-IR when adjusted for %BF or ACi;
Loprinzi et al., 2012 [42]	cross-sectional	535	ACC: n/r Volume (min/d) of MVPA	MS	Regression: adjusted for age, race and smoking (1) MVPA was favorably associated with the odds of being MS;
Macena et al., 2021 [43]	cross-sectional	58	ACC: ActivPAL; triaxial Sitting/lying down (1.25 METs), Standing (1.4 METs), Walking 120 steps/min (4 METs) (h/d) Steps/d	HOMA-IR	ANOVA: (1) Sitting/lying down, standing, walking, and steps/d had no association with HOMA-IR;
Panton et al., 2007 [44]	cross-sectional	35	PED: Yamax Digi-Walker SW-200, sealed Steps (n/d)	SBP, DBP; HbA1c, TC, HDL, LDL, TG, CRP	ANOVA: (1) Group (Steps/d < 5000) had lower TC, LDL compared to Group (Steps/d ≥ 5000); (2) No differences in SBP, DBP, HbA1c, HDL, TG, CRP between Groups.
Slater et al., 2021 [45]	cross-sectional	275	ACC: ActiGraph w-GT3X, Acti-Watch; triaxial Volume (min/d) of MVPA TPA (cpm/d)	HbA1c, FPG, HOMA-IR, TC, TG, HDL, LDL, SBP, DBP, Fasting insulin, CRP	Regression: adjusted for age, socioeconomic, %BF (1) In Group (Pacific), TPA was positivelyunfavorably associated with SBP; (2) In Group (European), TPA was unfavorably associated with HbA1c and CRP; (3) In Group (Pacific), MVPA was unfavorably associated with fasting insulin; (4) In Group (European), MVPA was favorably associated with HDL and HOMA-IR, unfavorably associated with fasting insulin and CRP; (5) In all, TPA was unfavorably associated with CRP and fasting insulin; MVPA was favorably associated with HOMA-IR and HDL, unfavorably associated with CRP and fasting insulin.
Tabozzi et al., 2020 [46]	cross-sectional	13	ACC: ActiGraph GT3X + BT; triaxial %Volume: LPA, MPA, VPA Volume (min/d) of MVPA Steps (n/d)	FPG, PPG	Regression: adjusted for age, SB (1) MVPA was favorably associated with peak PPG; Correlation: (1) %LPA, %MPA, %VPA, MVPA, steps had no association with FPG; (2)%MPA, %VPA, MVPA were negatively associated with PPG; (3)%LPA and Steps had no association with PPG;
Vella et al., 2011 [47]	cross-sectional	60	ACC: Actigraph GT1M; uniaxial meeting/not meeting MVPA (30 min/d)	FPG, Fasting insulin, HOMA-IR, TC, HDL, LDL, TG, CRP, SBP, DBP	*t-*test: (1) Meeting MVPA guidelines had favorable effects on TC and TG;
Vella et al., 2009 [48]	cross-sectional	60	ACC: Actigraph GT1M; uniaxial Steps (n/d)	FPG, HDL, TG, SBP, DBP	correlation: (1) Steps/d were favorably associated with FPG; Regression: adjusted for age, FFM, FM (1) Steps/d were favorably associated with HDL and TG;
Zając-Gawlak et al., 2017 [49]	cross-sectional	85	ACC: ActiGraph GT1M; uniaxial Steps (n/d)	MS	U test: (1) Group (Steps/d ≥ 12500) had lower number of MS criteria than Group (10,000–12,500) and Group(<10,000); (2) No differences in the number of MS between Group (10,000–12,500) and Group(<10,000); Odds ratios: Group (Steps/d ≥ 12500) had 3.84 times lower risk of being MS than Group (Steps/d < 12,500);

Note: ACC, accelerometer; ACi, abdominal circumference; ANOVA, analysis of variance; CRP, C-reactive protein; DBP, diastolic blood pressure; FFM, fat-free mass; FPG, fasting glucose; FM, fat mass; HbA1c, glycosylated hemoglobin; HDL, high-density lipoprotein; HOMA-IR, homeostasis model assessment of insulin resistance; IL-6, interleukin-6; LDL, low-density lipoprotein; LPA, low-intensity physical activity; METs, metabolic equivalents; MHO, metabolically healthy overweight/obese; MPA, moderate-intensity physical activity; MS, metabolic syndrome; MUO, metabolically unhealthy overweight/obese; MVPA, moderate-to-vigorous-intensity physical activity; PA, physical activity; PED, pedometer; PPG, postprandial glucose; SBP, systolic blood pressure; TC, total cholesterol; TG, triglyceride; TNF-alpha, tumor necrosis factor-α; TPA, total physical activity; U-test, Mann–Whitney U-test; VO_2peak_, peak oxygen uptake; VPA, vigorous physical activity.

**Table 2 biology-11-00925-t002:** Characteristics of intervention studies.

Reference	Study Design	Sample Size	Intervention	PA Measure	Health Outcome	Association
Hasan et al., 2018 [50]	Quasi-experimental design	52	9-week walking program asked to walk 10,000 steps per day	PED: KenzLifeCoder e-step Steps (n/d)	SBP, DBP, TC, TG, HDL, LDL, FPG, Fasting insulin, HOMA-IR, MS	*t*-test: (1) After intervention, LDL decreased; (2) In Group (18 ≤ BMI < 25), no intervention effect on cardiometabolic parameters; (3) In Group (BMI ≥ 25), after intervention, TG and fasting insulin decreased; (4) In Group (Steps/d > 7056), after intervention, TG decreased; (5) In Group (Steps/d < 7056), no intervention effect on cardiometabolic parameters; Correlation: (1) After intervention, steps/d were favorably associated with MS Score; (2) After intervention, in Group (18 ≤ BMI < 25), steps/d had no association with all parameters; (3) After intervention, in Group (BMI ≥ 25), steps/d were favorably associated with MS Score; unfavorably associated with SBP and DBP;
Hornbuckle et al., 2012 [51]	Random experiment	44	12-week exercise intervention Group 1: asked to walk 10,000 steps/d Group 2: asked to walk 10,000 steps/d + RT 2d/w	PED: New Lifestyles Digi-Walker SW-200 Steps (n/d)	SBP, DBP, HDL, TG, TC, HbA1c, CRP	ANOVA: (1) No changes in all parameters after intervention in Group 1; (2) HbA1c decreased after intervention in Group 2;
Moreau et al., 2001 [52]	Randomized controlled trial	24	24-week incremental walking program Group 1: 3 km increase in daily walking; CONT: maintain current physical activity	PED: Yamax SW200 pedometer Steps (n/d)	SBP, DBP, Fasting insulin, FPG, HOMA-IR	ANOVA: (1) SBP decreased after intervention in Group 1 compared with CONT; (2) No changes in other parameters in either group after intervention.
Musto et al., 2010 [53]	Quasi-experimental design	77	12-week incremental walking program; asked to increase steps/d by 10% per week; the progression was reduced to a 3% when steps/d reached 10,000 Group 1: improved steps/d by 3000 or greater; CONT: stopped participating or did not achieve step improvement level	PED: Sportline 330 Steps (n/d)	SBP, DBP, TG, FPG, HDL	ANOVA: (1) SBP and FPG decreased after intervention in Group 1;
Pal et al., 2011 [54]	Random experiment	28	12-week walking program; Group 1: asked to undertake 30 min of walking/day; with sealed pedometer Group B: asked to accumulate 10,000 steps/d, with unsealed pedometer	PED: Yamax Digi-Walker SW-200 Steps (n/d)	SBP, DBP	ANOVA: (1) No changes in SBP and DBP in either group after intervention.
Rodriguez-Hernandez et al., 2018 [55]	Crossover design study	10	3-condition multiple walking breaks Condition 1: 4-h SB; Condition 2: 4-h SB with 2-min of moderate-intensity walking every 30 min; Condition 3: 4-h SB with 5-min of moderate-intensity walking every 30 min.	ACC: ActiGraph GT3X; triaxial %Volume: LPA, MVPA	PPG, AUCglucose	ANOVA: (1) There were between-condition differences for both %LPA and %MVPA during experiment between all conditions; (2) There were between-condition differences for the 4h-PPG between Condition 1 and Condition 3; (3) No between-condition differences for 1 h-, 2 h-, and 3 h-PPG; (4) No between-condition differences for peak PPG; (5) 2h-AUCglucose was lower in Condition 3 compared to Condition 1;
Sugawara et al., 2006 [56]	Random experiment	17	12-week cycling training Group 1 (n = 8): 180–300 kcal/session, 3–5 sessions/week at 40% HRR Group 2 (n = 9): at 70% HRR	ACC: Lifecorder; uniaxial LPA (<4METs), MPA (4–6METs), VPA (>6METs) (min/d)	SBP, DBP	ANOVA: (1) No changes in SBP or DBP in eigher group after intervention.
Sugiura et al., 2002 [57]	Randomized controlled trial	27	24-month exercise intervention Group 1 (n = 14): 90-min exercise (40–60%VO_2max_) 1 d/w + asked to increase at least 2000–3000 steps/d CONT (n = 13): maintain current physical activity	PED: n/r Steps (n/d)	TC, HDL, TG, LDL	ANOVA: (1) TC decreased after intervention in Group 1; (2) HDL increased after intervention in Group 1 compared with CONT; Regression: age, BMI, menopausal status (1) Steps/d had no association with TC and HDL in Group 1 before intervention; (2) Steps/d were favorably associated with TC, HDL, ΔTC and ΔHDL in Group 1 after intervention;
Swartz et al., 2003 [58]	Quasi-experimental design	18	4-week control period followed by 8-week walking program	PED: Yamax Digi-Walker SW-200 Steps (n/d)	SBP, DBP, FPG, PPG, Fasting insulin, Postprandial insulin, HOMA-IR, AUCglucose, AUCinsulin	ANOVA: (1) SBP, DBP, 2 h-PPG, 2 h-AUC glucose decreased after intervention.

Note: ACC, accelerometer; ANCOVA, analysis of covariance; ANOVA, analysis of variance; AUC, the area under the curve; CONT, control; CRP, C-reactive protein; DBP, diastolic blood pressure; FPG, fasting glucose; HbA1c, glycosylated hemoglobin; HDL, high-density lipoprotein; HOMA-IR, homeostasis model assessment of insulin resistance; HRR, heart-rate reserve; LDL, low-density lipoprotein; LPA, low-intensity physical activity; METs, metabolic equivalents; MPA, moderate-intensity physical activity; MS, metabolic syndrome; MVPA, moderate-to-vigorous-intensity physical activity; PA, physical activity; PED, pedometer; PPG, postprandial glucose; SB: sedentary behavior; SBP, systolic blood pressure; TC, total cholesterol; TG, triglyceride; TPA, total physical activity; VO_2max_, maximal oxygen uptake; VPA, vigorous physical activity.

**Table 3 biology-11-00925-t003:** Characteristics of participants for included studies.

Reference	Country	Race	Sample Size	Age	Body Mass Index	Menstrual Status	Diet	Education	Lifestyle	Socio-Economic Level	Tobacco
Camhi et al., 2015 [36]	USA	African American (61%)	46	26.7 ± 4.7	31.1 ± 3.7	/	no affected medications and dietary supplements	/	/	/	non-smoker (80%)
Diniz et al., 2015 [37] meet PA guideline	Brazil	/	25	55.8 ± 7.2	26.9 ± 5.1	postmenopausal	no affected medications	/	physically active	/	/
Diniz et al., 2015 [37] not meet PA guideline	Brazil	/	24	61.6 ± 6.2	29.1 ± 9.0	postmenopausal	no affected medications	/	physically inactive	/	/
Graff et al., 2012 [38]	Brazil	Caucasian (73%)	68	28.0 ± 6.0	28.0 ± 6.0	premenopausal	no affected medications	/	/	/	/
Green et al., 2014 [39]	USA	Caucasian (92%)	50	24.0 ± 4.8	27.0 ± 4.8	premenopausal	no affected medications	collage (84%)	/	college student (84%)	no smoking for 6 months
Koniak-Griffin et al., 2014 [40]	USA	Latina	210	44.6 ± 7.9	32.6 ± 5.7	/	/	college or more (4%)	/	low income	/
Lecheminant et al., 2011 [41]	USA	Caucasian (90%)	264	40.1 ± 3.0	31.7 ± 6.9	premenopausal	/	college or more (50%)	/	/	non-smoker
Loprinzi et al., 2012 [42]	USA	Caucasian (73%)	535	49.3 ± 0.9	28.8 ± 0.3	/	/	/	/	/	non-smoker (60%)
Macena et al., 2021 [43]	Brazil	/	58	31.0 ± 7.0	33.3 ± 4.1	premenopausal	no affected medications	/	/	low income	/
Panton et al., 2007 [44]	USA	African American	35	48 ± 8	42.3 ± 9.8	/	no affected medications	/	/	low income	non-smoker (83%)
Slater et al., 2021 [45] Pacific normal	New zealand	Pacific	61	25.0 ± 7.0	25.9 ± 3.9	premenopausal	/	/	/	low income	/
Slater et al., 2021 [45] Pacific obesity	New zealand	Pacific	55	26.0 ± 6.0	35.6 ± 6.1	premenopausal	/	/	/	low income	/
Slater et al., 2021 [45] European normal	New zealand	European	85	30.0 ± 7.0	22.5 ± 2.1	premenopausal	/	/	/	less deprived	/
Slater et al., 2021 [45] European obesity	New zealand	European	74	33.0 ± 7.0	33.7 ± 3.8	premenopausal	/	/	/	less deprived	/
Tabozzi et al., 2020 [46]	Italy	/	13	32.5 ± 16.1	24.0 ± 3.3	/	no affected medications	/	physically inactive	university nurse students/research staff	/
Vella et al., 2011 [47] no meet PA Guideline	USA	Hispanic	42	25.2 ± 5.6	23.8 ± 4.0	/	no affected medications	/	/	/	no smoking for 6 months
Vella et al., 2011 [47] meet PA Guideline	USA	Hispanic	18	24.4 ± 4.9	23.0 ± 4.6	/	no affected medications	/	/	/	no smoking for 6 months
Vella et al., 2009 [48]	USA	Mexican and Mexican American	60	24.9 ± 0.7	23.6 ± 0.5	/	no affected medications	/	/	/	no smoking for 6 months
Zając-Gawlak et al., 2017 [49]	Poland	/	85	62.8 ± 5.9	27.6 ± 4.5	postmenopausal	/	/	physically active	the Third Age University student	/
Hasan et al., 2018 [50]	UAE	/	52	21.4 ± 4.8	27.5 ± 5.6	/	no affected medications	college	/	college student	/
Hornbuckle et al., 2012 [51]	USA	African American	44	49.0 ± 5.5	34.7 ± 6.4	/	/	/	physically inactive	/	no smoking for 6 months
Moreau et al., 2001 [52]	USA	/	24	54.0 ± 1.0	/	postmenopausal	/	/	physically inactive	/	non-smoker
Musto et al., 2010 [53] Control	USA	/	34	45.7 ± 9.5	29.5 ± 5.0	/	/	/	physically inactive	/	/
Musto et al., 2010 [53] Active	USA	/	43	46.3 ± 10.4	30.4 ± 5.5	/	/	/	physically inactive	/	/
Pal et al., 2011 [54] 10,000 steps	Australia	/	13	41.4 ± 2.7	28.9 ± 1.2	/	no affected medications	/	physically inactive	/	non-smoker
Pal et al., 2011 [54] 30 min walking	Australia	/	15	45.3 ± 2.2	29.7 ± 1.1	/	no affected medications	/	physically inactive	/	non-smoker
Rodriguez-Hernandez et al., 2018 [55]	USA	/	10	36.0 ± 5.0	38.0 ± 1.6	/	no affected medications	/	physically inactive	/	/
Sugawara et al., 2006 [56] moderate intensity training	Japan	Asian	8	58.0 ± 4.0	25.5 ± 3.6	postmenopausal	/	/	physically inactive	/	non-smoker
Sugawara et al., 2006 [56] vigorou intensity training	Japan	Asian	9	59.0 ± 6.0	24.2 ± 3.0	/	/	/	/	/	/
Sugiura et al., 2002 [57] intervention	Japan	Asian	14	48.6 ± 4.2	22.3 ± 1.6	both	no affected medications	/	physically inactive	/	/
Sugiura et al., 2002 [57] control	Japan	Asian	13	48.0 ± 3.6	22.6 ± 1.9	both	no affected medications	/	physically inactive	/	/
Swartz et al., 2003 [58]	USA	/	18	53.3 ± 7.0	35.0 ± 5.1	both	/	/	physically inactive	/	non-smoker

**Table 4 biology-11-00925-t004:** The meta-analysis of the association between physical activity and cardiometabolic health outcomes (all analyses were performed using the random-effect model).

Study Group	No. Studies	Meta-Analysis	*p*	Heterogeneity	
Variables		r (95%CI)		I² (%)	*p*
MVPA (min/day)					
DBP	3	0.04 (−0.06, 0.13)	0.47	15	0.31
SBP	3	0.01 (−0.08, 0.09)	0.83	0	0.53
FPG	3	−0.09 (−0.19, 0)	0.06	10	0.33
HDL	3	0.16 (0.06, 0.25)	0.002	19	0.29
LDL	3	0.06 (−0.02, 0.15)	0.15	0	0.53
TC	3	0.09 (0, 0.18)	0.04	0	0.4
TG	3	−0.02 (−0.15, 0.11)	0.74	49	0.14
Steps/day					
Glucose	3	−0.12 (−0.24, 0.01)	0.06	0	0.48
HDL	4	0.24 (−0.07, 0.49)	0.13	81	0.001

Note: DBP, diastolic blood pressure; FPG, fasting glucose; HDL, high-density lipoprotein; LDL, low-density lipoprotein; MVPA, moderate-to-vigorous-intensity physical activity; SBP, systolic blood pressure; TC, total cholesterol; TG, triglyceride.

**Table 5 biology-11-00925-t005:** The meta-analysis of the effect of meeting physical activity guideline on HOMA-IR (all analyses were performed using the random-effect model).

Variables	No. Studies	Meta-Analysis		Heterogeneity	
HOMA-IR		SMD (95%CI)	*p*	I² (%)	*p*
Meet vs. Not meet	3	−0.22 (−0.46, 0.02)	0.08	11	0.32

Note: HOMA-IR, homeostasis model assessment of insulin resistance.

## Data Availability

Not applicable.

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
