# Peer review of "Associations between Objectively Determined Physical Activity and Cardiometabolic Health in Adult Women: A Systematic Review and Meta-Analysis"

_biology, 2022, doi:10.3390/biology11060925_

Round 1

Reviewer 1 Report

I believe the authors tap into an important topic in health the relationship between physical activity and cardiometabolic health indicators.

This study aims to analyze the relationship between cardiorespiratory fitness, adiposity, and cardiometabolic risk in adult women populations. I think some things need clarifying for the publication that will help in the overall interpretation and understanding of the results before being published within the scope of the Journal.

Title:  This systematic review was registered in PROSPERO system with code PROSPERO 2022 CRD42022307774. I suggest to the authors change the title to “Physical activity and health indicator in adult women: a systematic review and meta-analysis:”

Please delete page 2.

Introduction

Comment 1: The manuscript is well written and clear.

Comment 2: (line 82) – Please add this information: Https://www.crd.york.ac.uk/prospero/display_record.php?RecordID=307774

Comment 3: (line 72) – The authors identified 10217 documents, but 1927 are duplicates. 1927 manuscripts identified as "duplicates" were removed. Subsequently, the authors selected 3145 manuscripts for analysis, but 10217 total, minus 1927 duplicates, results in 8290 manuscripts. Can the authors clarify this fact?  

Comment 4: Figure 1 should be improved.

Comment 5: Please uniformize the type of letter (i.e.: Reference - Study design - Sample size - PA measure - Health Outcome – Association)

Comment 6: Please identify the abbreviation PA.

Comment 7: To better understand the data, improve the forest plots.

Author Response

Dear Reviewer,

All the line numbers in my last response corresponded to the manuscript displayed in all-visible “Track Changes” in Microsoft Office Word. In consideration of the fact that you may have received a PDF version with all track changes invisible, we have changed all line numbers in this new response. Sorry for the inconvenience and hope to meet with the final approval.

Comments:

  1. This systematic review was registered in PROSPERO system with code PROSPERO 2022 CRD42022307774. I suggest to the authors change the title to “Physical activity and health indicator in adult women: a systematic review and meta-analysis:”

Response: Thank you very much for your comments. We added “a systematic review and meta-analysis”, the revised title is Associations between Objectively Determined Physical Activity and Cardiometabolic Health in Adult Women: A Systematic review and Meta-analysis.

  1. Please delete page 2.

Response: Thank you very much for your comments. We had removed it.

  1. line 82: – Please add this information: Https://www.crd.york.ac.uk/prospero/display_record.php?RecordID=307774

Response: Thank you very much for your comments. We added the website as in Line 779.

  1. line 72 – The authors identified 10217 documents, but 1927 are duplicates. 1927 manuscripts identified as "duplicates" were removed. Subsequently, the authors selected 3145 manuscripts for analysis, but 10217 total, minus 1927 duplicates, results in 8290 manuscripts. Can the authors clarify this fact?  

Response: Thank you very much for your comments. A total of 5105 records were identified from the database search between 1st January 1990 and 31st January 2022 from PubMed, Scopus, Web of Science, and the Cochrane Library. Additionally, 7 records were identified through reference list screening. After removing 1967 duplicate records, 3145 studies were further screened based on title and abstract. Exclusion of irrelevant studies resulted in 126 records. After examining the full text, finally, 23 eligible studies were included in the present review.

  1. Figure 1 should be improved.

Response: Thank you very much for your comments. Figure 1 had been improved. “Qualitative synthesis” was changed to “systematic review”. Additionally, we added details of sample size of varying study types.

  1. Please uniformize the type of letter (i.e.: Reference - Study design - Sample size - PA measure - Health Outcome – Association)

Response: Thank you very much for your comments. The type of letter had been uniformized.

  1. Please identify the abbreviation PA.

Response: Thank you very much for your comments. All the abbreviations in the context had been identified.

  1. To better understand the data, improve the forest plots.

Response: Thank you very much for your comments. We improved the forest plots. The improved ones sorted by the type of physical activity and it was consistent with the statement.

 Once again, thank you very much for your suggestions and comments, and we feel highly honored by your support

Reviewer 2 Report

This study evaluated the association and interactions between various objective measures of physical activity and various measures of cardiometabolic risk among women. This study's contribution synthesizes existing literature on these factors in these populations. Strengths of this study include a meta-analysis of associations among diverse populations (up to 8 countries, unclear) represented in included studies. Overall, this article will be acceptable for publication addressing the comments discussed below.

See attached document for detailed review.

Author Response

Dear  Reviewer,

All the line numbers in my last response corresponded to the manuscript displayed in all-visible “Track Changes” in Microsoft Office Word. In consideration of the fact that you may have received a PDF version with all track changes invisible, we have changed all line numbers in this new response. Sorry for the inconvenience and hope to meet with the final approval.

Comments for introduction:

  1. Remove blank page 2?

Response: Thank you very much for your comments. We have removed it.

  1. Some of the referenced papers are from a younger population than was included in this study and these differences are not discussed in light of the current project.

Response: Thank you very much for your comments. We changed these references to those more relevant as in Line 57. (Katzmarzyk et al., 2010; Ekelund et al., 2019)

  1. Line 50-52: Awkward sentence, reword or split into two sentences.

Response: Thank you very much for your comments. We reworded the sentence to make it clearer as in Lines 57-58. Physical inactivity has been epidemiologically evidenced to be associated with cardiovascular diseases (CVD), which remains the leading cause of mortality [Katzmarzyk et al., 2010; Ekelund et al.,2019]. Since physical inactivity is among the major modifiable risk factors for CVD, there is a growing need for the promotion of physical activity (PA).

  1. Line 60: “for other patterns”: provide a brief example or discussion of what these are.

Response: Thank you very much for your comments. We added a brief example as in Lines 68-69. Other patterns include light intensity physical activity, total volume of physical activity and the number of daily steps.

  1. Line 68-70: behavioral and socioeconomic differences between men and women are germane to the topic and not included in the discussion not or in this analysis.

Response: Thank you very much for your comments. We removed this sentence because examining the differences between men and women is not the purpose of the current review.

  1. As this is a multi-country sample, differences in PA guidelines, at least among included countries, should be discussed or it should be stated all are the same. Considering the depth of the topic and the numerous included measures and associations evaluated, the introduction is lacking details and clear discussion of gaps in the literature and the need to synthesize existing literature.

Response: Thank you very much for your comments. Basically, 2020 Worth Health Organization guideline on physical activity and 2018 physical activity guideline for Americans are the most used basis for setting national guidelines. These guidelines focus on the time spent on PA performed at least moderate intensity. To our knowledge, no guideline specifies the amount of light intensity PA, total PA, or daily steps. We add the description as in Lines 68-69. Additionally, we rewrite the discussion about the research gap as in Lines 78-79.

Comments for methods:

  1. The use of bullet points does not enhance readability and understanding of the selection process.

Response: Thank you very much for your comments. We change it to 2.1.1. Participants as you suggested.

  1. Clarify the types of observational studies and intervention.

Response: Thank you very much for your comments. We specify the types of studies included as in Lines 124-125. Both observational (cross-sectional and longitudinal) and experimental (randomized and non-randomized) were included.

  1. In the event that two different manuscripts from the same study each met one of the criteria, how was selection made?

Response: Thank you very much for your comments. We clarify the inclusion criteria as in Line 135-137. When more than one article was from the same study, the following hierarchy was applied for inclusion: 1) the largest sample size, 2) the longest following period, and 3) the most detailed data.

  1. U-test/ K-S test; there are non-standard abbreviations.

Response: Thank you very much for your comments. We use the full name for these words at the first time as in Line 156, and then use the abbreviation in the following context.

  1. It appears separate NOS scales used for cohort and case-control studies (by maximum number of stars)? Please clarify. Which Cochrane collaboration toll was used?

Response: Thank you very much for your comments. According to a previous systematic review [Ramsey et al., 2022], the high versus low quality was defined as a score greater or equal to the median of total possible points. The total stars a longitudinal study can be awarded is 9 and for a cross-sectional study, the maximal stars were 7. Therefore, the cut-offs were < or ≥ 4 for cross-sectional studies and < or ≥ 5 for longitudinal studies. Additionally, the Cochrane risk of bias assessment tool was used [Higgins et al., 2011]

Comments for statistical analysis:

  1. Line 173: Suggest “To evaluate the impact of heterogeneity on the meta-analysis, inconsistency was measured using I2

Response: Thank you very much for your comments. We changed the sentence as you suggested.

  1. Line 173-174: Please clarify which percentage RANGE applies to which category.

Response: Thank you very much for your comments. We specified the Higgins’ index of I2 as in Lines 193-194. To evaluate the impact of heterogeneity on the meta-analysis, inconsistency was measured using the Higgins’ I2 statistic. Specifically, I2 = 0 indicated no heterogeneity and the low, moderate, and high heterogeneity was identified when I2 < 25%, 25-75%, and > 75% respectively.

  1. Line 174-175: What were the 10 studies used to assess publication bias with Egger’s test and funnel plots?

Response: Thank you very much for your comments. According to the Cochrane Handbook, tests for funnel plot asymmetry should be used only when there are at least 10 studies included in the meta-analysis, because when there are fewer studies the power of the tests is too low to distinguish chance from real asymmetry. Therefore, we set at least 10 studies.

  1. Line 175-177: How were subgroup analyses conducted? Provide more details.

Response: Thank you very much for your comments. We added details for subgroup as in Lines 198-199.

 Comments for results:

  1. Sample characteristics is hard to follow. Suggest including a table to delineate sample characteristics.

Response: Thank you very much for your comments. We added Table 3 for sample characteristics.

  1. Line 215: youngest included age is listed as 18y, but this indicates including 5 studies with women before menarche, please clarify.

Response: Thank you very much for your comments. Sorry for confusing you. It is a type error. We have corrected it to postmenopausal and premenopausal as in Lines 264-265.

  1. Line 227: correct Saw to See

Response: Thank you very much for your comments. We have corrected Saw to See as in Line 278.

  1. PA guidelines were set at 150 min moderate/week, is this the same for all countries included in this analysis? Why was this chosen (explain in Methods)

Response: Thank you very much for your comments. At the beginning, it was not our purpose to assess the effect of the adherence of the PA guidelines. Therefore, whether meet or not meet PA guideline is not included in the searching key word. When we summarized the evidence, we found three articles reporting this variable, so we presented it in the results. However, since we lack the knowledge of PA guideline for every country included, we can't say that PA guideline in every country is the same. Therefore, 150 minutes of moderate intensity PA, the most used recommendation by WHO, was chosen as the standard for pooled analysis.

  1. Line 235-237: “categorized using various” dose this statement apply to methods employed in this study or methods employed in the sample studies? Please clarify.

Response: Thank you very much for your comments. Due to the different device and cut-points standard, in the section of method, we stated that LPA, MPA, VPA, MVPA and TPA were defined as reported in the studies. If studies measured physical activity in metabolic equivalent tasks (METs), we used the cut-points proposed Ainsworth et al. (2011) (e.g. 1.6–2.9 METs was defined as LPA, 3-5.9 METs as MPA, and ≥6 METs as VPA) as in Lines 177-181.

  1. Line 246: consider your evaluation of IL-6, it may be pro-inflammatory or anti-inflammatory (in cases of acute PA when released from muscle)

Response: Thank you very much for your comments. We included this indicator as clinical data affirm the relationship between IL-6 signaling and cardiovascular events [Ferrante and Condorelli et al., 2019]. Furthermore, it is a cytokine with a complex signaling pattern and is becoming clinically actionable as a target in atherosclerosis therapy [Libby et al, 2021].

  1. Line 252-253: What age selected as the most important criteria a priori or post hoc.

Response: Thank you very much for your comments. Age was selected as the most important criteria a priori as described in the Methods section in Line 161.

  1. Association between PA and cardiometabolic health outcomes – a clearer presentation and rewording would benefit the reader.

Response: Thank you very much for your comments. We rewrote this section as in Lines 322-443. Findings from different study design were presented and discussed.

  1. RCT should be singular for “One”

Response: Thank you very much for your comments. We corrected it.

  1. Lines 291-292: “unfavorable relationship” implies increased PA leads to poorer LDL and TC, clarify.

Response: Thank you very much for your comments. Unfavorable relationship between MVPA and TC was reported in overweight Latin women (Koniak-Griffin et al. 2014). Another higher level of LDL and TC was reported in more active obese African American women with lower socioeconomic status (Panton et al., 2007). It is hard to explain. As authors stated, one explanation is the measurement error from motion sensors and another explanation is that a great deal more PA is need for obese women to lower clinical indicators. Furthermore, TC and LDL may be more affected by other factors such as dietary habits, however, it was not assessed in studies included. Further research is needed.

  1. Line 296: clarify “rather different findings”.

Response: Thank you very much for your comments. The association between MVPA and lipid outcomes were highly inconsistent in observational cross-sectional studies. As we rewrote the sentences, such confusing words are removed. Sorry about that.

  1. Lines 294-301: What type of studies are discussed in this paragraph? Clarify.

Response: Thank you very much for your comments. This paragraph is about the findings from observational cross-sectional studies. As we rewrote the sentences, such confusing paragraph has been removed. Sorry about that.

  1. Lines 324-325: Is this a summary of studies discussed? Unclear

Response: Thank you very much for your comments. We rewrote these sentences as in Lines 379-389.

  1. Lines 400-402: include sample size for pooled and subgroup analysis

Response: Thank you very much for your comments. We added the sample size.

  1. Figure 1: Suggest replacing “qualitative analysis” with “systematic review”.

Response: Thank you very much for your comments. We changed it to “systematic review” as in Figure 1.

  1. Suggesting adding details of sample size of varying study types as indicated in Lines 189-193.

Response: Thank you very much for your comments. We specified study types as in Figure 1.

  1. (a) The relationship between__and__, these appear to be part of a new table set.

Response: Thank you very much for your comments. We presented the summarized data for meta-analysis in Table 4 and visual forest plot in Figure 2.

Comments for discussion:

  1. Line 477: add “the” – “suggested by most of the studies”

Response: Thank you very much for your comments. We added it.

  1. Line 490-492: What does the evidence suggest in terms of types of RCT and prospective studies should evaluate? Please be more specific

Response: Thank you very much for your comments. We specified the evidence as in Lines 600-604. In addition to walking intensity, evidence from experimental studies indicated that the baseline value of biomarkers and magnitude of changes on daily steps affected the relationship between walking and cardiometabolic health outcomes. Likewise, body composition variables suggested by observational cross-sectional evidence could also mediate this relationship.

  1. Lines 513: Explain VO2peak and how it relates, consider including introduction

Response: Thank you very much for your comments. We added the description of VO2peak as in Lines 633-644. It was evidenced by previous cross-sectional studies that VO2peak was associated with risk factors of cardiovascular disease (CVD) with a moderate to strong correlations (Abdulnour et al., 2010). Likewise, Kodama et al. (2009) conducted a meta-analysis to quantitatively define the relationship between cardiorespiratory fitness and the incidence of CVD. The authors indicated that those with low cardiorespiratory fitness had a risk ratio for CVD events of 1.56 compared to those with high cardiorespiratory fitness (Kodama et al., 2009). Therefore, the cardiorespiratory fitness appeared to be an important confounder when investigating the relationship between PA and cardiometabolic health. Furthermore, the cardiorespiratory fitness should be taken into consideration when developing exercise protocols aiming to improve cardiometabolic health. Since high intensity exercises were well documented to be effective and efficient on improving cardiorespiratory fitness (Sultana et al., 2019; O’ Donoghue et al., 2021; Kong et al., 2016; Wen et al., 2019), in this regard, PA performed at higher intensity was recommended.

  1. Lines 538-551: What is the authors’ evaluation of this review in light of individual findings? Please elaborate.

Response: Thank you very much for your comments. We added some evidence from prospective and cross-sectional studies and the discussion of our evaluation as in Lines 686-692. These results were in agree with findings from prospective studies that short spurts of MVPA could provide protection against the onset of hypertension (White et al., 2015) and all-cause mortality (Saint-Maurice et al., 2018). Although MVPA in 10 min bouts were generally recommended for health benefits, the accumulated evidence from cross-sectional and prospective studies showed that short-lived MVPA was associated with health outcomes. As such, the move towards recommending MVPA of any bout duration through the lasted PA guideline tended to be a pragmatic change (Piercy et al., 2018; Bull et al., 2020).

  1. Lines 552-561: Taken together, what do these findings indicate? Add summary statement.

Response: Thank you very much for your comments. We added summary statement as in Lines 703-707. Despite that most evidence was from cross-sectional analysis; it was encouraging that the promotion of short bouts MVPA was more likely to be feasible for most women. From the public health perspective, it had significant implications for inactive individuals as health benefits could be achieved by simply being more active without emphasizing the exercise duration.

  1. Line 580: delete “answer”, suggest “further clarify or something similar”

Response: Thank you very much for your comments. We changed the word “answer” to “further clarify” as in Line 734.

  1. Lines 588-589: address synthesis of very low and low quality in the discussion, compare low-high quality evidence presented.

Response: Thank you very much for your comments. We added the statement as in Lines 743-745.

  1. Line 594: were statistical methods applied to control for confounding variables in the included studies? Verify. Causal inferences can be suggested through Hill’s criteria or epidemiological web of causation which include dose-response.

Response: Thank you very much for your comments. We added the statement as in Lines 747-756. Secondly, most of the included studies were cross-sectional in design, using t-test or ANOVA without controlling for any potential confounders such as age and body composition. An initiative to address this issue was setting age to be the most import confounder in rating the quality of studies included. Furthermore, the most adjusted data were included in the discussion and meta-analysis. In addition, although sedentary behavior was documented to be associated with cardiometabolic health, it was not assessed in the current review. Taken together, the absence of these confounders attenuated the association between PA and cardiometabolic health, and our findings should be interpreted with caution.

Comments for conclusion:

  1. Line 613, 616-617: statements are partially contradictory, rewrite or clarify writing.

Response: Thank you very much for your comments. We corrected it as in Line 800. Findings from the present systematic review and meta-analysis provide evidence that PA is not associated with most cardiometabolic health outcomes in healthy adult women. However, it is most compelling that being more physically active is beneficial for the MS. For women, it makes more sense to emphasize the volume of PA rather than whether the volume of PA is accumulated by bouts or is sporadic.

 Once again, thank you very much for your suggestions and comments, and we feel highly honored by your support

Reviewer 3 Report

The present manuscript aimed to review and analyze the associations between objectively measured physical activity and cardiometabolic health in adult women. The topic of the study is very interesting, and the authors did a great effort to achieve the aim of the review. However, the great, perhaps excessive, heterogeneity of studies considered makes very difficult to follow the manuscript and to get proper conclusions. Therefore, in my opinion, the main problem with the present review is to find common points among studies considered to allow significant results and conclusions.

In spite of some of the following points are reported again as specific comments, and also some of them have been acknowledged as limitations of the study, the authors should focus their attention to the following main general question. The authors should clarify and explain properly how they have managed together:

-Measures of physical activity performed using such different devices (pedometers and accelerometers), when even measures performed using accelerometers from different brands are not consistent, use different algorithms, give different parameters, etc.? I don’t know the reasons the authors consider this a strength of the study.

-Results from observational and interventional, and even quasi-experimental, studies?

-Studies with participants of different ethnicities?

-A lot of different cardiometabolic markers, from blood pressure to HOMA-IR and to metabolic syndrome (lines 240-247)? Again, in my opinion, and in order to increase in clarity, this is not a strength of the study. More parameters allowing the inclusion of more studies in the review can not be a reason to consider this a strength. This could even lead to consider the present review as a group of different reviews, for example when contents in lines 262-348 are analyzed or when pooled analysis are performed with, for example, data from only six studies for meta-analysis (line 350) or four studies for daily steps and HDL cholesterol (line 387). In this regard, each parameter could be also more influenced by different characteristics of physical activity, mainly intensity or duration.

-Several variables out of control in all studies?

-Observational and interventional studies? The authors consider this a strength of the study, but I don’t agree with this.

-Studies where participants presented at least two cardiovascular risk factors (refs 51, 52 and others) when other where this condition seems to be unknown? Actually, this supposes that studies with specific interventions focused on people with a significant cardiovascular risk were considered together with other observational studies.

Specific comments

Abstract and keywords

-Line 41. Please note that the last recommendations are at least 150-300 minutes per week. In spite of maybe it could be simplified to “at least 150 minutes”, this does not describe the exact recommendations. This will be commented again.

-In my opinion “objectively” can not be considered as a keyword.

-It would be better to join women and adult to make a more informative keyword such as “adult women”.

Introduction

-Lines 52-54. Actually, the recommendation is "at least 150– 300 minutes of moderate-intensity aerobic physical activity; or at least 75–150 minutes of vigorous- intensity aerobic physical activity; or an equivalent combination of moderate- and vigorous-intensity activity throughout the week". Therefore, it is not "at least 150 minutes...". Therefore, the common "minimum 150 minutes" has been modified. Even, it is indicated that time longer than 300 minutes should be considered to get additional benefits. This could be also considered when discussing the main findings of the study.

-Lines 66-68. This is a key point in the background to justify the review. Therefor, in my opinion, this specific features should be described, or at least listed, in more detail. This could help to a better understanding of review development and to ensure that the authors have achieved the aims.

Methods

-No inclusion or exclusion criteria regarding the number of participants is considered. In this regard, most of the studies reported a low, or a very low (n=13-18 for observational and 10 for a cross-over design). I think that this question requires a comment, even considering that PA was measured objectively rather than simply with questionnaires.

-Line 88. This is too vague and no informative at all. It should be removed. In my opinion this is a key point in the study. Actually, studies including only obese women are considered in the review. Which is the relevance of this criterion when studies focused on only obese or when high SBP is reported for all these participants (lines 442-443). Which studies have been excluded taking into account this criterion?

-Line 110. It should clarify how interventional studies were considered, e.g. pre or post-intervention data, both, etc. and how this was combined with studies of a different nature such as the observational ones.

-Line 123. I understand that the lack of proper studies is the main reason to consider such old manuscripts (1990). Otherwise, it seems that too old studies are considered for this review.

-TNF-a is not correct. Please correct, here and elsewhere, this typographical, or style, mistake to TNF-a (-alpha).

Results and discussion

-Did the authors find any effect of age?

-Did the authors find any influence of menopausal status?

-The possible influence of different ethnicities on results observed should be also commented as, for example, cardiovascular risk as well as other diseases, are different and present different prevalence among different ethnicities.

-Lines 344-348. Which definition for metabolic syndrome was used? Can the authors report the key parameter leading to the positive effect on the MS score? Actually, within the present review, it would be more consistent reporting parameters involved in the MS definition rather than the effect or association with overall MS.

-Line 317. Considering recent findings, this statement (“replacing sitting…”) highlights one of the main limitations of this review, and of the studies included. Sedentarism has not been considered. This, together with listing other potential confounders, should be added in lines 593-594.

-Lines 601-604. In relation with this, it would be good to include in the manuscript a brief discussion about the differences between objectively and subjectively measured PA, e.g. high or low correlations, level of correlations for different PA intensities, etc. This could help to understand some results from the review.

Conclusions

-Line 613. In my opinion, this is a too specific conclusion for this review. As it has been indicated above, I’m not sure about the convenience of using a result of an algorithm, as it is MS, in the present review together with many other single markers.

-Lines 618-619. As it has been indicated above, this could be confusing taking into account the lowest range recommended in the last review of the recommendations.

-Lines 620-622. In my opinion, and taking into account all the limitations of the review, such important and general suggestion should be removed. Rather than this, maybe the last conclusion should be that more studies, and more homogenous are needed to properly complete this review.

Author Response

Dear Reviewer,

All the line numbers in my last response corresponded to the manuscript displayed in all-visible “Track Changes” in Microsoft Office Word. In consideration of the fact that you may have received a PDF version with all track changes invisible, we have changed all line numbers in this new response. Sorry for the inconvenience and hope to meet with the final approval.

General Comments:

  1. Measures of physical activity performed using such different devices (pedometers and accelerometers), when even measures performed using accelerometers from different brands are not consistent, use different algorithms, give different parameters, etc.? I don’t know the reasons the authors consider this a strength of the study.

Response: Thank you very much for your comments. Most previous studies and physical activity (PA) guidelines mostly relied on the subjective measures such as questionnaire and interview. These subjective measures are difficult to provide precise measures of light intensity physical activity (LPA) (Skender et al., 2016). Although LPA accounts for the large of our daily physical activity, it received much less research attention. Furthermore, subjective measures are less accurate in women [Ferrari et al., 2007], it may be due to the fact that women spend more time in light intensity physical activity. Others, in the last decade it has been recognized that PA represents a continuum. Objective measures can provide us a more comprehensive knowledge of the relationship between PA and health. Therefore, there is a need to use objectively determined PA in investigating the relationship with cardiometabolic health in women.

To deal with different devices, we analysis data from pedometers and accelerometers separately. PA measures from pedometers were identified as steps and PA measures from accelerometers were discussed as total physical activity and PA at different intensities, such as LPA, moderate physical activity (MPA), moderate-to-vigorous intensity physical activity (MVPA) and vigorous physical activity (VPA). However, there is no standard cut-points for accelerometer studies. According to previous systematic review and meta-analysis published in good journal (Ramakrishnan et al.,2021), LPA, MPA, MVPA, VPA are defined as reported in the study. We initially included all data and different cut-points were presented in the supplementary document. Overall, the cut-points used include Freedson et al. (1998), Sasaki et al. (2011), Troiano et al. (2008) and Matthews et al. (2005). Among them, the Matthews’s cut-point for MVPA of 760 counts per minutes was much lower than others, which is used in only one study included in our review (Camhi et al., 2005). This study is not included in the pooled analysis. Nonetheless, other discrepancies exist, and we descript it as a limitation in Lines 757-769 that findings much be interpreted with caution.

  1. Results from observational and interventional, and even quasi-experimental, studies?

Response: Thank you very much for your comments. In the result section, we present findings for each health outcome by study type. Likewise, in the section of discussion, we initially discuss findings from different study types and compared them to high quality evidence from other references. We try to explain the differences if there are inconsistent results.

  1. Studies with participants of different ethnicities?

Response: Thank you very much for your comments. Most studies included do not report ethnicity of participants. We add characteristics of participants as in Table 3. Furthermore, we conduct a subgroup analysis for studies with different ethnicities. We also specify the inconsistent results from different ethnic groups in the section of discussion.

  1. A lot of different cardiometabolic markers, from blood pressure to HOMA-IR and to metabolic syndrome (lines 240-247)? Again, in my opinion, and in order to increase in clarity, this is not a strength of the study. More parameters allowing the inclusion of more studies in the review can not be a reason to consider this a strength. This could even lead to consider the present review as a group of different reviews, for example when contents in lines 262-348 are analyzed or when pooled analysis are performed with, for example, data from only six studies for meta-analysis (line 350) or four studies for daily steps and HDL cholesterol (line 387). In this regard, each parameter could be also more influenced by different characteristics of physical activity, mainly intensity or duration.

Response: Thank you very much for your comments. We remove “a wide range of cardiometabolic health indicators” as a strength in this review since the small number of studies included in the meta-analysis. However, the association between MVPA and most health indicators is pooled and analysis. As MVPA is emphasized for public health, it can provide a comprehensive knowledge that how MVPA is associated with different cardiometabolic health outcomes. Meanwhile, the associations with steps, TPA, LPA, MPA are discussed. It is important and encouraging for individuals who seek the primary prevention from PA as they could determine the most feasible, acceptable, and effective way to increase PA and health.

  1. Several variables out of control in all studies?

Response: Thank you very much for your comments. We presented statistical method in Table 1 and 2. Among observational studies, 7 studies used t-test or ANOVA do not control for any variables. The other 7 studies used regression, correlation or ANCOVA control for variables, including age, SB, MVPA, VO2peak, BM, FM, FFM, %BF, race, smoking and socioeconomic status. We correct the statement in the limitation as in Line 747-756.

  1. Observational and interventional studies? The authors consider this a strength of the study, but I don’t agree with this.

Response: Thank you very much for your comments. Including a wide range of study types allow us to compare low with high quality evidence. We separately discussed findings from observational and interventional studies and find some differences. For example, evidence from cross-sectional studies reveal that steps are not associated with blood pressure. However, three studies conducted a long-term walking program in obese women and observed decreases in SBP after the intervention (Moreau et al., 2001; Musto et al., 2010; Swartz et al., 2003). The discrepancy is due to the baseline value and the magnitude of increase in PA.

  1. Studies where participants presented at least two cardiovascular risk factors (refs 51, 52 and others) when other where this condition seems to be unknown? Actually, this supposes that studies with specific interventions focused on people with a significant cardiovascular risk were considered together with other observational studies.

Response: Thank you very much for your comments. Our aim is to examine the relationship between PA and cardiometabolic health in apparently health women. We try to explore a general relationship for public health. Although we cannot confirm the causal relationship between PA and health indicators, we find some correlations through comparison indeed. For example, HDL is significantly correlated with PA, while other indicators are not. In addition, although some indicators are not significantly correlated with PA, the indicator improved following interventions. For example, in cross-sectional analysis, blood pressure is not associated with daily steps. However, systolic blood pressure is improved by increasing steps after walking programs. We analysis the discrepancy and find that the baseline value of indicator and the changes of steps affect the relationship. This finding provides support for those having risk factors for improving their health through PA.

Specific comments

Abstract and keywords

  1. Line 41. Please note that the last recommendations are at least 150-300 minutes per week. In spite of maybe it could be simplified to “at least 150 minutes”, this does not describe the exact recommendations. This will be commented again.

Response: Thank you very much for your comments. Data included is only compared on the threshold of 150 minutes moderate intensity physical activity, we change the wording as in Lines 45-47 to emphasize the minimal level of recommendation. 

  1. In my opinion “objectively” can not be considered as a keyword.

Response: Thank you very much for your comments. We remove the keyword “objectively”

  1. It would be better to join women and adult to make a more informative keyword such as “adult women”.

Response: Thank you very much for your comments. We change the keyword to “adult women”.

Introduction

  1. Lines 52-54. Actually, the recommendation is "at least 150– 300 minutes of moderate-intensity aerobic physical activity; or at least 75–150 minutes of vigorous- intensity aerobic physical activity; or an equivalent combination of moderate- and vigorous-intensity activity throughout the week". Therefore, it is not "at least 150 minutes...". Therefore, the common "minimum 150 minutes" has been modified. Even, it is indicated that time longer than 300 minutes should be considered to get additional benefits. This could be also considered when discussing the main findings of the study.

Response: Thank you very much for your comments. We change “at least 150 minutes; 75minutes” to the range “at least 150-300 minutes; 75-150 minutes” as in Line 60.

  1. Lines 66-68. This is a key point in the background to justify the review. Therefor, in my opinion, this specific features should be described, or at least listed, in more detail. This could help to a better understanding of review development and to ensure that the authors have achieved the aims.

Response: Thank you very much for your comments. We add the description about specific features as in Lines 77-80. For example, women suffer the higher age-related risk of hypertension than man (Saeed et al., 2017). Furthermore, physical inactivity in women is more likely to be diagnosed with obesity (Vainshelboim et al., 2019; Cooper et al., 2021).

Methods

  1. No inclusion or exclusion criteria regarding the number of participants is considered. In this regard, most of the studies reported a low, or a very low (n=13-18 for observational and 10 for a cross-over design). I think that this question requires a comment, even considering that PA was measured objectively rather than simply with questionnaires.

Response: Thank you very much for your comments. Because our inclusion criteria are already rigorous, as we require separate data on healthy women, it limits the number of eligible studies. The most common reason for exclusion is the absence of data on women. In order to include more studies, there is no requirement on the number of samples in individual study. However, we include the number of samples in the evaluation of evidence quality. If the total number of samples is less than 400, the quality grade will be lowered by one level. Additionally, we added it as one of the limitations as in Lines 742-743.

  1. Line 88. This is too vague and no informative at all. It should be removed. In my opinion this is a key point in the study. Actually, studies including only obese women are considered in the review. Which is the relevance of this criterion when studies focused on only obese or when high SBP is reported for all these participants (lines 442-443). Which studies have been excluded taking into account this criterion?

Response: Thank you very much for your comments. We rewrote the inclusion criteria of participant to make it clearer as in Lines 103-109. The criteria are made as a prior. Apparently healthy women with a mean age of 18-64 years. Women with the presence of cardiovascular disease risk factors (e.g., overweight/obesity, hypertension, elevated fasting glucose, and dyslipidemia) were included. Exclusion criteria included: 1) diagnosed CVD, diabetes, physical or psychological disorders, or other conditions that were barrier to physical activities; 2) pregnant, postpartum, or lactating women; 3) elite athletes.

  1. Line 110. It should clarify how interventional studies were considered, e.g. pre or post-intervention data, both, etc. and how this was combined with studies of a different nature such as the observational ones.

Response: Thank you very much for your comments. We clarify the study type included in the correlational meta-analysis as in Lines 185-186. Additionally, we clarify how to calculate the standardized mean difference for meta-analysis as in Lines 511-512. Unfortunately, we did not conduct the standardized mean difference meta-analysis for intervention studies in this review due to the high heterogeneity in exercise protocols.

  1. Line 123. I understand that the lack of proper studies is the main reason to consider such old manuscripts (1990). Otherwise, it seems that too old studies are considered for this review.

Response: Thank you very much for your comments. We would like to include more studies if they are proper and eligible. However, we find that there has been very little research focused on women all the time. It is exactly the current research gap. We expected more attention for women as their PA level is lower than man and are more easily to suffer from physical inactivity.

  1. TNF-a is not correct. Please correct, here and elsewhere, this typographical, or style, mistake to TNF-a (-alpha).

Response: Thank you very much for your comments. We correct it to TNF-alpha.

Results and discussion

  1. Did the authors find any effect of age?

Response: Thank you very much for your comments. The effect of age is presented as in Lines 488-489. The pooled association between MVPA and high-density lipoprotein (HDL) is not significant in young women. We also find a significant association between steps and HDL in middle-aged women and studies conducted in Japan as in Lines 505-506. Meanwhile, we indicate that results must be interpreted with caution because the number of studies included in the subgroup is small.

  1. Did the authors find any influence of menopausal status?

Response: Thank you very much for your comments. The effect of menopausal status is presented as in Lines 488-490. The pooled association between MVPA and high-density lipoprotein (HDL) is not significant in premenopausal women.

  1. The possible influence of different ethnicities on results observed should be also commented as, for example, cardiovascular risk as well as other diseases, are different and present different prevalence among different ethnicities.

Response: Thank you very much for your comments. We described ethnicity related difference in the relationship as in Line 490, Line 516, and Lines 617-621. As examining the ethnicity related difference is not our purpose in the present review and there is small number of studies included in the subgroup, we add it as a limitation in the current review.

  1. Lines 344-348. Which definition for metabolic syndrome was used? Can the authors report the key parameter leading to the positive effect on the MS score? Actually, within the present review, it would be more consistent reporting parameters involved in the MS definition rather than the effect or association with overall MS.

Response: Thank you very much for your comments. We specified different definitions of MS score for each study as in Line 589, Line 593, and Lines 663-666.

  1. Line 317. Considering recent findings, this statement (“replacing sitting…”) highlights one of the main limitations of this review, and of the studies included. Sedentarism has not been considered. This, together with listing other potential confounders, should be added in lines 593-594.

Response: Thank you very much for your comments. We add it as the limitation in Line 752-756.

  1. Lines 601-604. In relation with this, it would be good to include in the manuscript a brief discussion about the differences between objectively and subjectively measured PA, e.g. high or low correlations, level of correlations for different PA intensities, etc. This could help to understand some results from the review.

Response: Thank you very much for your comments. We add the difference between objectively and subjectively measured PA as in Lines 725-734. In spite of limitations of our relative low-quality evidence, the results of the current review showed some support that objectively measured PA was not beneficial to cardiometabolic outcomes. However, the majority of studies using subjectively determined PA consistently reported a favorable association with health outcomes (Schultz et al., 2020; Pitanga et al., 2019; Crichton et al., 2015). This discrepancy was mainly due to the weak correlation between subjective and objective methods for assessing the intensity and duration of PA (Nascimento-Ferreira et al., 2018). We were unable to judge which one was superior because both had several limitations. Therefore, a combination of subjective and objective methods would be expected to further clarify some of the issues revealed by this study.

Conclusions

  1. Line 613. In my opinion, this is a too specific conclusion for this review. As it has been indicated above, I’m not sure about the convenience of using a result of an algorithm, as it is MS, in the present review together with many other single markers.

Response: Thank you very much for your comments. We correct the conclusion as in Line 800. Findings from the present systematic review and meta-analysis provide evidence that objectively measured PA is not associated with most cardiometabolic health outcomes in healthy adult women.

  1. Lines 618-619. As it has been indicated above, this could be confusing taking into account the lowest range recommended in the last review of the recommendations.

Response: Thank you very much for your comments. We change the words as in Lines 807-808. The present review also highlights that meeting 150 minutes of MVPA weekly recommended is scarcely enough to observe significant beneficial effects.

  1. Lines 620-622. In my opinion, and taking into account all the limitations of the review, such important and general suggestion should be removed. Rather than this, maybe the last conclusion should be that more studies, and more homogenous are needed to properly complete this review.

Response: Thank you very much for your comments. We remove this sentence and rewrite it as in Lines 809-811. However, further high-quality studies with less heterogeneity are still needed to yield compelling findings of the association between PA and cardiometabolic health.

Once again, thank you very much for your suggestions and comments, and we feel highly honored by your support

Reviewer 4 Report

Lu et. al. perform in their paper “Associations between objectively determined physical activity volumes and cardiometabolic health in adult women” a meta-analysis of multiple studies to analyze the potential of physical activity to improve cardiometabolic health in women. The meta study follows Preferred Reporting Items for Systematic Reviews and Meta-Analyses standards. The rationale of the analysis, dataset inclusion and exclusion are clearly described in the paper. Due to their strict criteria and the narrow group investigated, most of the screened studies had to be excluded, so that the final result, that physical activity did not lead to an improvement of the investigated metabolic makers except HDL, is only based on a small number of included studies. This small number is further complicated due to the heterogenicity of those. However, the authors clearly point out those limitations and discuss them extensively. Although the study might not be able to discern too many novel findings, it clearly points to the need of more, extensive studies in the field and should therefore be published. 

Author Response

Dear Reviewer,

Thank you very much for your suggestion. Requiring data from women exclusively has led to a sharp decline in eligible studies. It indicates that there has been little research focusing on women. However, women tend to be less physically active and are more vulnerable to physical inactivity. We add the conclusion that “However, further high-quality studies with less heterogeneity are still needed to yield compelling findings of the association between PA and cardiometabolic health in women.”

 Once again, thank you very much for your suggestions and comments, and we feel highly honored by your support

Reviewer 5 Report

Dear authors,

After reading the manuscript " Associations between objectively determined physical activity volumes and cardiometabolic health in adult women", I believe there can be an interesting contribution for the current state of the art in this specific topic. The clarity and flow in some parts still need improvement. Hence, I've recommended revision to improve further text clarity before I can consider recommending it for publishing, according to the following reasons:

Title: please include in the title: a systematic review and meta-analysis.

Keywords: Objectively - I suggest replacing this word by other more related to the work.

Cardiometabolic; Adult - please avoid repeating the words that are present in the title

There is an extra page (p2). Please remove it

Introduction

L55 – ref [3] à I suggest including a reference from ACSM as well

L67-68 - Please add specific anatomical, hormonal, and cardiovascular features differences between men and women

L68-70 - Please explain what are those special behavioral and socioeconomic characteristics

L72-76 - It is suggested providing some examples for subjectively measures of PA and objective measures, for better clarity

Methods

L 71 - Please rewritten the whole article according the most recent PRISMA guidelines (2020).

L91-95 - In the previous sentence, it is mentioned, apparently healthy and asymptomatic women but this sentence refers the presence of some disease and risk factors. Please clarify.

L97-100 - how was they measured? Were these devices considered for objectively measuring PA? Did you analyse the validity and reliability of these devices? How was light, moderate, vigorous defined? Please explain.

Section 2.4 - The description of the research is much more complete in the supplementary table A. My suggestion is to include it in the manuscript.

Table 2 - please increase the size of the column so the word “sample” can fit in the same line

It is suggested to include Tables C and D from supplementary file in the manuscript for better clarity

Text in the results section should avoid starting sentences with numbers.

Please avoid starting sentences with of. Instead, you can use “From”.

Discussion is very well written. My suggestion is to develop a little more the section 4.5. For instance, future direction suggestions are missing.

L606-607 - In my perspective this can't be considered a limitation because it was addressed as an exclusion criteria.

Conclusion

Please add practical applications and examples derived from this study regarding the type of PA (I assume that was all from walking, but please clarify because there are other types as well), time, volume and the level of intensity considering different ages, pre, post or menopausal, and BMI. Try to make a link with the benefits that were found.   

L614-616 - This sentence is not clear. Expressions such amount of PA and total volume need to be clarified over the text. Please consider my previous comments.

Nonetheless, this is a relevant work and a good contribution to the state of art.

Thank you

Best regards

Author Response

Dear Reviewer,

All the line numbers in my last response corresponded to the manuscript displayed in all-visible “Track Changes” in Microsoft Office Word. In consideration of the fact that you may have received a PDF version with all track changes invisible, we have changed all line numbers in this new response. Sorry for the inconvenience and hope to meet with the final approval.

Comments:

  1. Title: please include in the title: a systematic review and meta-analysis.

Response: Thank you very much for your comments. We include it and the title now is: Associations between Objectively Determined Physical Activity and Cardiometabolic Health in Adult Women: A Systematic review and Meta-analysis

  1. Keywords: Objectively - I suggest replacing this word by other more related to the work.

Response: Thank you very much for your comments. We remove the keyword “objectively” and add the keyword “physical activity”.

  1. Cardiometabolic; Adult - please avoid repeating the words that are present in the title.

Response: Thank you very much for your comments. We change the keyword to: Accelerometer; Pedometer; Physical activity; Steps; Cardiometabolic health; Adult women

  1. There is an extra page (p2). Please remove it

Response: Thank you very much for your comments. We have removed the blank page.

Introduction

  1. L55 – ref [3] à I suggest including a reference from ACSM as well.

Response: Thank you very much for your comments. We add it. Moreover, we change the 150 minutes of MPA to 150-300 minutes and 75 minutes of VPA to 75-150 minutes to make it clearer as in Line 60.

  1. L67-68 - Please add specific anatomical, hormonal, and cardiovascular features differences between men and women

Response: Thank you very much for your comments. We add it as in Line XX. For example, women had smaller size of vessel than men and suffer the higher age-related risk of hypertension, especially among postmenopausal with decreased estrogen [Saeed et al., 2017]. Furthermore, physical inactivity in women is more likely to be diagnosed with obesity [Vainshelboim et al., 2019; Cooper et al., 2021].

  1. L68-70 - Please explain what are those special behavioral and socioeconomic characteristics

Response: Thank you very much for your comments. Those special behaviors and socioeconomic characteristics include women do more house working, low income, low education level and so on. However, we remove it in the revised version as these physical activity patterns and characteristics are not assessed and discussed in the present review.

  1. L72-76 - It is suggested providing some examples for subjectively measures of PA and objective measures, for better clarity

Response: Thank you very much for your comments. We rewrite this sentence as in Lines 84-86. Furthermore, previous studies and PA guidelines mostly relied on the PA questionnaire, which is less accurate in women [Ferrari et al., 2007] and is difficult to provide precise measures of LPA [Skender et al., 2016].

Methods

  1. L 71 - Please rewritten the whole article according the most recent PRISMA guidelines (2020).

Response: Thank you very much for your comments. We move the registration information to a new section following the discussion. Furthermore, we add a brief summary and limitation at the beginning of the discussion. Other sections seem to comply with PRISMA (2020) statement.

  1. L91-95 - In the previous sentence, it is mentioned, apparently healthy and asymptomatic women but this sentence refers the presence of some disease and risk factors. Please clarify.

Response: Thank you very much for your comments. We tended to exclude women with any conditions that are barriers to physical activity. We rewrote the section of participant as in Lines 103-109. Apparently healthy women with a mean age of 18-64 years. Women with the presence of cardiovascular disease risk factors (e.g., overweight/obesity, hypertension, elevated fasting glucose, and dyslipidemia) were included. Exclusion criteria included: 1) diagnosed CVD, diabetes, physical or psychological disorders, or other conditions that were barrier to physical activities; 2) pregnant, postpartum, or lactating women; 3) elite athletes.

  1. L97-100 - how was they measured? Were these devices considered for objectively measuring PA? Did you analyse the validity and reliability of these devices? How was light, moderate, vigorous defined? Please explain.

Response: Thank you very much for your comments. Accelerometer and pedometer measured PA were considered objective. The validity and reliability of the device were reported in the study. If the study did not report the validity and reliability, we downgrade the quality of the study. LPA, MPA, MVPA, VPA and TPA were defined as reported in the study. Since the heterogeneity between studies using different devices, we added it as the limitation in Lines 757-769 and stated that these findings must be interpreted with caution.

  1. Section 2.4 - The description of the research is much more complete in the supplementary table A. My suggestion is to include it in the manuscript.

Response: Thank you very much for your comments. We add the search strategy in the Appendix A in the manuscript.

  1. Table 2 - please increase the size of the column so the word “sample” can fit in the same line

Response: Thank you very much for your comments. This issue had been fixed in the revised version.

  1. It is suggested to include Tables C and D from supplementary file in the manuscript for better clarity

Response: Thank you very much for your comments. Supplementary Table C was included in the main text Table 3 and Supplementary Table D was included in the Appendix B.

  1. Text in the results section should avoid starting sentences with numbers.

Response: Thank you very much for your comments. We had corrected these errors.

  1. Please avoid starting sentences with of. Instead, you can use “From”.

Response: Thank you very much for your comments. We had changed it to “From”.

  1. Discussion is very well written. My suggestion is to develop a little more the section

4.5. For instance, future direction suggestions are missing.

Response: Thank you very much for your comments. We added implications for practice and future research as in Lines 782-797. Our systematic review and meta-analysis found that accelerometer and pedometer derived PA were not associated with most individual cardiometabolic health outcomes. These findings were inconsistent with those based on the subjectively measured PA. Improvement in objective measures in the future including the gender-specific cut-points, activity pattern recognition was more likely to improve our knowledge of the health benefits of PA.

Our review found the evidence that walking program was effective in increasing daily steps among adult women, while significant improvements in cardiometabolic indicators were hardly observed following interventions, except among obesity participants. However, some improvements on SBP were reported among obese women with higher SBP value at baseline. Furthermore, we found that increasing PA was associated with higher HDL, however, such favorable association was attenuated among young women. Further research should pay greater attention to potential confounders such as age, body composition and cardiorespiratory fitness when investigating the association between PA and cardiometabolic health in adult women.

  1. L606-607 - In my perspective this can't be considered a limitation because it was addressed as an exclusion criteria.

Response: Thank you very much for your comments. We removed it.

Conclusion

  1. Please add practical applications and examples derived from this study regarding the type of PA (I assume that was all from walking, but please clarify because there are other types as well), time, volume and the level of intensity considering different ages, pre, post or menopausal, and BMI. Try to make a link with the benefits that were found. 

 Response: Thank you very much for your comments. We add implications for practice and future research as in Lines 782-797.

  1. L614-616 - This sentence is not clear. Expressions such amount of PA and total volume need to be clarified over the text. Please consider my previous comments.

Response: Thank you very much for your comments. Sorry for the typing error. We had corrected it as in Line 800.

Once again, thank you very much for your suggestions and comments, and we feel highly honored by your support

Round 2

Reviewer 3 Report

The manuscript has been largely improved. However, in  my opinion, a too high degree of variability of markers, measures... have been considered. I don't know whether reasons reported by authors are enough to explain this excessive variability wich, in turn, account for a low level of he manucript, albeit a larger amount of contents.

The version of the manuscript I have received presents a lot of "format" mistakes. For example:

-Lines 2-4. Title together with authors.

-Incompelte Figure 1

-Line 299 (next subsection title appears in this line)

-Lines 699-707, inclcuding the conclusion section title

-Line 717

...

On the ther hand it seems that there are some, or a lot of, mistakes due to changes introduced. This should bre reviewed carefully. For example, and only regarding to a previous comment:

"TNF-alphalpha" (appendix A and Table 1), TNF-a, tnf-α, tumor necrosis factor-a (line 278)...

-Please, consider also an improvement in the quality of figures, mainly text in FIgures 2 and 3.

-A last comment. I'm not used to see acknowledgements in the first person singular. It seems that this comes from only one author. Takint into account the colective nature of the work, and the presence of seven authors, I think this is not common or, even, correct.

Author Response

Dear Reviewer,

In consideration of the fact that you may have received a version with errors induced by format change, we have uploaded a PDF version in this new response. Sorry for the inconvenience and hope to meet with the final approval.

Comments: The version of the manuscript I have received presents a lot of "format" mistakes. For example:

-Lines 2-4. Title together with authors.

Responses: Thank you very much for your comments. The title and authors have been separated in the revised version. This error may be due to the format changes. Sorry about that.

-Incompelte Figure 1

Responses: Thank you very much for your comments. Figure 1 had been presented completely as in Line 208.

-Line 299 (next subsection title appears in this line)

Responses: Thank you very much for your comments. The subsection title 3.6.1 Blood pressure has been removed to a new line.

-Lines 699-707, inclcuding the conclusion section title

Responses: Thank you very much for your comments. This paragraph has been separated.

-Line 717

...

On the ther hand it seems that there are some, or a lot of, mistakes due to changes introduced. This should bre reviewed carefully. For example, and only regarding to a previous comment:

"TNF-alphalpha" (appendix A and Table 1), TNF-a, tnf-α, tumor necrosis factor-a (line 278)...

Responses: Thank you very much for your comments. All abbreviations have been checked.

-Please, consider also an improvement in the quality of figures, mainly text in FIgures 2 and 3.

Responses: Thank you very much for your comments. The forest plot was automatically generated by Review Manager. Correlational and SMD were presented separately. We grouped the forest plot according to MVPA and Step to make it clearer. Zooming in a little bit might make the picture clearer. Meanwhile we provided a summary table as in Table 4 and Table 5.

-A last comment. I'm not used to see acknowledgements in the first person singular. It seems that this comes from only one author. Takint into account the colective nature of the work, and the presence of seven authors, I think this is not common or, even, correct.

Responses: Thank you very much for your comments. We rewrote the acknowledgement and added author contributions.

Acknowledgments: We would like to thank Yaodong Gu from Faculty of Sport Science at Ningbo University for feedback on the manuscript at all stages of this study.

Author Contributions: Yining L. was a major contributor to conceptualizing the systematic review. Yining L., Q.W., and S.Y. participated in screening and extracting data. Yining L., W.H., and J.B. assessed methodological quality. J.L. and Yichen Lu. were responsible for visualization. Yining L. drafting and writing the manuscript. J.B. and W.H. revised the manuscript and provided feedback on the content and structure of the manuscript. All authors had read and agreed to the published version of the manuscript.

Once again, thank you very much for your suggestions and comments, and we feel highly honored by your support!